# Plenodium: Underwater 3D Scene Reconstruction with Plenoptic Medium Representation

**Changguang Wu**[1]   **Jiangxin Dong**[1*]   **Chengjian Li**[1]   **Jinhui Tang**[2*]

[1]Nanjing University of Science and Technology, [2]Nanjing Forestry University

{changguangwu, jxdong, lichengjian}@njust.edu.cn, tangjh@njfu.edu.cn

https://plenodium.github.io/

## Abstract

We present *Plenodium* (*plenoptic medium*), an effective and efficient 3D representation framework capable of jointly modeling both objects and the participating medium. In contrast to existing medium representations that rely solely on view-dependent modeling, our novel plenoptic medium representation incorporates both directional and positional information through spherical harmonics encoding, enabling highly accurate underwater scene reconstruction. To address the initialization challenge in degraded underwater environments, we propose the pseudo-depth Gaussian complementation to augment COLMAP-derived point clouds with robust depth priors. In addition, a depth ranking regularized loss is developed to optimize the geometry of the scene and improve the ordinal consistency of the depth maps. Extensive experiments on real-world underwater datasets demonstrate that our method achieves significant improvements in 3D reconstruction. Furthermore, we construct a simulated dataset with GT and the controllable scattering medium to demonstrate the restoration capability of our method in underwater scenarios.

## 1 Introduction

Underwater imaging plays a critical role in diverse applications, including underwater construction, marine sciences, etc. However, its efficacy is significantly hindered by the complex optical properties of the aquatic environment. These properties lead to wavelength- and distance-dependent attenuation and scattering of light, resulting in degraded image quality characterized by the color cast, diminished contrast, and loss of detail. Given the expanding scientific and industrial focus on oceanic exploration, the reconstruction of scattering-affected underwater scenes becomes increasingly important.

Pioneering works [3–7] based on Neural Radiance Fields (NeRF) [8] and 3D Gaussian Splatting (3DGS) [9] have significant contributions to 3D reconstruction. These methods achieve effective surface modeling by constraining their representations to scene surfaces, assuming a vacuum-like medium between the observer and the objects. However, such approaches neglect the influence of light scattering in participating media, limiting their applicability in real underwater environments.

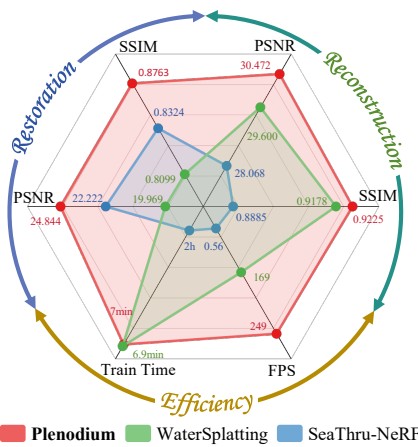

Figure 1: Comparison of Plenodium, WaterSplatting [1], and SeaThru-NeRF [2] on reconstruction and restoration performance (PSNR, SSIM), as well as efficiency (FPS, training time).

---

*Jiangxin Dong and Jinhui Tang are corresponding authors.

39th Conference on Neural Information Processing Systems (NeurIPS 2025).

To achieve underwater 3D scene reconstruction, SeaThru-NeRF [2] extends the NeRF framework by introducing an additional medium volume field, which is parameterized by an MLP and characterizes the color and density attributes of the medium, enabling accurate modeling of light-medium interactions. WaterSplatting [1] leverages 3DGS as an alternative geometric representation to replace the computational NeRF in SeaThru-NeRF [2], while preserving the core medium representation capability, achieving more efficient rendering. However, there are three significant limitations affecting its accuracy, efficiency, and robustness: 1) Existing methods estimate medium properties based exclusively on viewing directions, failing to account for the spatial relationship between camera positions and scattering effects in heterogeneous media; 2) Despite its advantages, WaterSplatting still employs an implicit MLP for medium representation, which introduces substantial computational costs limiting its efficiency; 3) The 3DGS-based approaches rely on COLMAP [10, 11] for initializing Gaussian primitives, but underwater image degradation severely impairs COLMAP's feature extraction and matching, compromising the reliability of the initialization process.

In this paper, we present an effective and efficient method for underwater 3D reconstruction. Different from existing methods that rely on the medium representation with limited directional information, we are the first to take positional information into account and develop a plenoptic medium representation. Notably, the proposed plenoptic medium representation is modeled by explicit Spherical Harmonics (SH), rather than implicit MLPs. Specifically, we positionally encode the SH coefficients via a trilinear interpolation mechanism to capture accurate scattering effects in arbitrary positions, while achieving faster rendering than MLP-based methods. Then, to address the limitations of COLMAP in degraded underwater scenes, we propose a pseudo-depth Gaussian complementation method that enriches the sparse point clouds with pseudo-depth estimated from the Depth Anything Model [12, 13], improving the robustness of the initialization for 3DGS. Furthermore, we introduce a depth ranking regularized loss to optimize the geometry of the scene, enhancing the ordinal stability of the depth maps. Taken together, the proposed approach can effectively improve the reconstruction quality while speeding up rendering. In addition, we created a simulated dataset for validating the restoration performance of our approach across various types of media and different degradation intensities, as well as analyzing degradation impacts on 3D reconstruction.

The contributions can be summarized as follows: 1) We propose *Plenodium*, which introduces a novel plenoptic medium representation that characterizes both the directional and positional information and then incorporates it with 3DGS for effective underwater 3D reconstruction. 2) To improve the robustness of 3DGS-based reconstruction in underwater scenarios, we introduce a pseudo-depth Gaussian complementation to enrich COLMAP-initialized Gaussian primitives and a depth ranking regularized loss to enhance the geometric consistency. 3) We construct a simulated dataset with ground truth (GT) and a controllable scattering medium, which enables systematic evaluation of image restoration performance across degradation levels. 4) Extensive experiments demonstrate the effectiveness and efficiency of our approach. As shown in Fig. 1, Plenodium outperforms prior methods, increasing the PSNR by at least 0.872dB and speeding up the rendering efficiency by 47% in real-world reconstruction scenarios.

## 2 Related Work

**3D Gaussian splatting**. 3DGS [9] constructs a 3D scene representation with a set of 3D Gaussians, where the $i$-th Gaussian is defined by a center position $\mu_i \in \mathbb{R}^3$, a 3D covariance matrix $\Sigma_i \in \mathbb{R}^{3 \times 3}$, an opacity $\sigma_i \in \mathbb{R}$, and color features $A_i$. Specifically, the rendered color $\hat{C}$ is computed by a blending process that combines the color contributions $\{\hat{C}_i\}_{i=1}^N$ from $N$ individual Gaussians:

$$\hat{C} = \sum_{i=1}^N \hat{C}_i = \sum_{i=1}^N c_i \alpha_i T_i, \text{ where } T_i = \prod_{j=1}^{i-1}(1 - \alpha_j), \tag{1}$$

where Gaussian color $c_i = \text{SH}(d, A_i)$ is derived from spherical harmonics [14] with ray direction $d$ and its color features $A_i$, and $\alpha_i$ is computed by multiplying its $\sigma_i$ and its projected 2D Gaussian.

3DGS addresses limitations of the reconstruction efficiency in 3D scene modeling by leveraging explicit 3D Gaussian primitives for real-time rendering [5, 6] and minute-level training [15, 16] on consumer GPUs. 3DGS has been proven to be effective in a wide range of applications, including digital human reconstruction [17, 18], Artificial Intelligence Generated Content [19–21], and autonomous driving [22, 23]. In computational imaging, 3DGS has also shown significant improvements

in super-resolution [24], deblurring [25, 26], derain [27], and low-light enhancement [28]. These advancements demonstrate the capacity of 3DGS to invert ill-posed imaging problems, transforming low-quality inputs into high-quality 3D reconstructions [29–33] with photometric consistency.

**Computer vision with scattering medium**. Underwater computer vision faces significant challenges due to complex optical phenomena, particularly light scattering and wavelength-dependent attenuation. These effects degrade image quality by introducing color distortion, reduced contrast, and haze, rendering traditional computer vision methods (designed for clear-air environments) ineffective in underwater applications. To address the ill-posed nature of the problem, earlier work introduces domain-specific priors to restore the scenes [34–37]. The method in [38] proposes a general image formation model in scattering media under ambient illumination, expressing per-pixel color $C$ as:

$$C = \underbrace{c^{obj} \cdot (e^{-\sigma^{att} \cdot z})}_{\text{direct}} + \underbrace{c^{med} \cdot (1 - e^{-\sigma^{bs} \cdot z})}_{\text{backscatter}}, \tag{2}$$

where $c^{obj}$ denotes the intrinsic color of the object at depth $z$, $c^{med}$ represents the ambient medium color at infinite distance, $\sigma^{att}$ and $\sigma^{bs}$ are the attenuation coefficients for the direct and backscatter components. Building on this framework, SeaThru [39] leverages depth maps to decouple attenuation and backscatter estimation, achieving robust restoration by explicitly modeling depth-dependent light propagation. Osmosis [40] adopts an unsupervised diffusion framework that iteratively refines images using priors derived from unpaired clean and degraded datasets. Furthermore, Seathru-NeRF [2] and WaterSplatting [1] extend NeRF and 3DGS architectures by embedding the physical model into their rendering equations, enabling simultaneous 3D scene reconstruction and water removal.

## 3 Preliminaries

In contrast to vanilla 3DGS, our Plenodium incorporates explicit modeling of medium-induced light attenuation through absorption and scattering effects, following [1]. The transmittance $T_i(z)$ at a given depth $z$ along the ray, situated between the $(i\text{-}1)$-th and $i$-th Gaussian splat is formulated as:

$$T_i(z) = T^{med}(z) \cdot T_i^{obj}, \tag{3}$$

where $T^{med}(z) = e^{-\sigma^{med} z}$ represents the exponential attenuation due to medium absorption, characterized by the medium's extinction coefficient $\sigma^{med}$ along the path from the camera to depth $z$, and $T_i^{obj} = \prod_{j=1}^{i-1}(1 - \alpha_j)$ captures the cumulative transmittance through all preceding Gaussian primitives, quantifying their occlusion effects on downstream geometry.

Meanwhile, the medium's contribution to color $\hat{C}_i^{med}$ between these Gaussian splats is computed as:

$$\hat{C}_i^{med} = \int_{z_{i-1}}^{z_i} c^{med} \sigma^{med} T_i^{obj} T^{med}(z) \, dz = c^{med} T_i^{obj} (e^{-\sigma^{med} z_{i-1}} - e^{-\sigma^{med} z_i}), \tag{4}$$

where $z_i$ denotes the depth of the $i$-th Gaussian in the camera coordinate system ($z_0$ is set to 0).

Following [1, 2], we utilize two sets of parameters: object attenuation $\sigma^{att}$ for object color $\hat{C}^{obj}$ and medium backscatter $\sigma^{bs}$ for medium color $\hat{C}^{med}$. The comprehensive rendering equation is:

$$\hat{C} = \hat{C}^{obj} + \hat{C}^{med} = \sum_{i=1}^{N} \hat{C}_i^{obj} + \sum_{i=1}^{N} \hat{C}_i^{med} + \hat{C}_\infty^{med}$$

$$= \sum_{i=1}^{N} c_i \alpha_i T_i^{obj} e^{-\sigma^{att} z_i} + \sum_{i=1}^{N} c^{med} T_i^{obj} (e^{-\sigma^{bs} z_{i-1}} - e^{-\sigma^{bs} z_i}) + c^{med} T_{N+1}^{obj} e^{-\sigma^{bs} z_N}. \tag{5}$$

where $\hat{C}_\infty^{med}$ presents the medium's color contribution from the last Gaussian to infinitely far.

## 4 Plenodium

We aim to develop an efficient and robust 3D reconstruction method for underwater scenarios. Specifically, our approach begins with an explicit plenoptic medium representation (*i.e.*, Fig. 2(b))

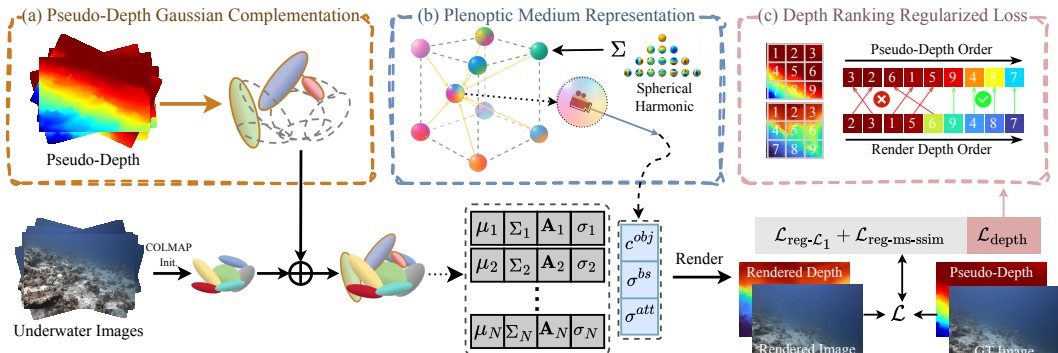

Figure 2: Overview of our *Plenodium*. We first employ the pseudo-depth Gaussian complementation to enrich the primitives initialized by COLMAP. Then we utilize the plenoptic medium representation to estimate the medium parameter and render the underwater images following Eqn. 5. The primitives are optimized with the loss function in Eqn. 12, including a new depth ranking regularized loss.

to accurately model the scattering medium in Sec. 4.1. We then present a pseudo-depth Gaussian complementation (*i.e.*, Fig. 2(a)) that leverages robust depth priors to enhance Gaussian primitive initialization in Sec. 4.2. Finally, a depth ranking regularized loss (*i.e.*, Fig. 2(c)) is introduced to enhance the geometric consistency in Sec. 4.3. The whole framework is summarized in Fig. 2.

## 4.1 Plenoptic Medium Representation

Scattering effects in heterogeneous media exhibit complex dependencies on both ray direction $d$ and the observer's spatial coordinates $(x, y, z)$. While existing methods only account for directional dependency while neglecting positional variations, we propose a novel plenoptic medium representation that explicitly incorporates both directional and positional information. Specifically, we parameterize the properties of the medium along each ray (*i.e.*, the medium color $c^{med}$, the object attenuation coefficient $\sigma^{att}$, and the medium backscatter coefficient $\sigma^{bs}$ in Eqn. 5) by Spherical Harmonics (SH):

$$c^{med} = \text{SH}(d, A_{x,y,z}^{c^{med}}), \quad \sigma^{att} = \text{SH}(d, A_{x,y,z}^{\sigma^{att}}), \quad \sigma^{bs} = \text{SH}(d, A_{x,y,z}^{\sigma^{bs}}), \qquad (6)$$

where $A_{x,y,z}^{c^{med}}, A_{x,y,z}^{\sigma^{att}}, A_{x,y,z}^{\sigma^{bs}}$ denote the SH coefficients for medium color, object attenuation, and medium backscatter at position $(x, y, z)$, respectively.

To enable efficient learning of spatially varying medium parameters from sparse camera observations, we store eight fundamental SH coefficients at each corner of the normalized 3D space $[-1, 1]^3$. Through trilinear interpolation [41] of these, we can reconstruct the SH coefficient set at any arbitrary spatial position:

$$A_{x,y,z}^t = \frac{1}{8} \sum_{u,v,w \in \{-1,1\}} (1 + u \cdot x)(1 + v \cdot y)(1 + w \cdot z) A_{u,v,w}^t, \quad \text{where } t \in \{c^{med}, \sigma^{att}, \sigma^{bs}\}. \quad (7)$$

The proposed plenoptic medium representation employs explicit SH encoding [14], reducing the time required for medium parameter retrieval during inference to less than 5% of that needed by implicit MLP-based methods [1, 2], while simultaneously improving the accuracy of scattering simulations in heterogeneous media (as illustrated in Sec. 6).

## 4.2 Pseudo-Depth Gaussian Complementation

Underwater 3DGS initialization is challenged by severe light attenuation and scattering, which degrade conventional SfM pipelines [10] like COLMAP [11]. To address this, we propose Pseudo-Depth Gaussian Complementation (PDGC), which builds on COLMAP by inheriting its initial Gaussians and augmenting them with additional ones guided by monocular pseudo-depth priors.

First, for a given camera view, we render the pixel-wise depth $\hat{D}$ via $\alpha$-blending according to:

$$\hat{D} = \sum_{i=1}^{N} (z_i \alpha_i \prod_{j=1}^{i-1} (1 - \alpha_j)) / (1 - T_{N+1}^{obj}), \qquad (8)$$

where $T_{N+1}^{obj}$ represents the accumulated object transmission behind all $N$ Gaussians.

Then, we estimate the corresponding pseudo-depth map $\tilde{D}$ using the RGB image $C$ under this camera view with the Depth Anything Model [12, 13], chosen for its superior generalization ability across diverse scenarios. To address the scale ambiguity and offset biases in monocular depth estimation, we formulate an affine correction to calibrate the pseudo-depth:

$$\tilde{D}' = k\tilde{D} + b, \quad \text{where } k, b = \arg\min_{k,b} \sum_{(x,y)\in\Omega_w} \left(\hat{D}(x,y) - k\tilde{D}(x,y) - b\right)^2. \quad (9)$$

The $k$ and $b$ are optimized via least-squares over well-initialized regions $\Omega_w = \{(x,y)|T_{N+1}^{obj}(x,y) < \tau_w\}$, refining $\tilde{D}$ to reduce scale ambiguity and offset, and aligning it with the initialized scene.

Subsequently, we determine the region $\Omega_n \cap \Omega_p$ to insert new Gaussian primitives, specifically targeting regions exhibiting both proximal camera distance ($\Omega_n = \{(x,y)|\tilde{D}(x,y) < \tau_{near} \cdot \max(\tilde{D})\}$) and elevated transmittance values ($\Omega_p = \{(x,y)|T_{N+1}^{obj}(x,y) \geq \tau_w\}$), which minimizes background interference and oversampling, thus improving efficiency and reconstruction accuracy.

Finally, we add new Gaussian primitives at each pixel $(x,y)$ in the defined region $\Omega_n \cap \Omega_p$ with depth $\tilde{D}'(x,y)$. *More details on the Gaussian primitive insertion process and the other attributes of the inserted Gaussian primitives can be found in the Sec. A.2.*

Our pseudo-depth Gaussian complementation method enriches the initialized Gaussian primitives across diverse scenes (Tab. 4), improving the robustness of 3DGS initialization against degradations.

## 4.3 Loss Function

Building upon differentiable 3DGS frameworks, we develop a multi-objective optimization pipeline for primitive refinement, jointly enforcing photometric accuracy, structural coherence, and depth consistency. Following [1, 42], we incorporate a weighting matrix $W = \frac{1}{\text{sg}(\hat{C})+\epsilon}$ (with $\epsilon = 10^{-6}$, $\text{sg}(\cdot)$ means stop gradient) to emphasize dark regions during optimization, aligning with human perceptual sensitivity to dynamic range. Then, based on the L1 loss and the multi-scale differentiable SSIM loss [43], we employ two regularized losses:

$$\mathcal{L}_{\text{reg-}\mathcal{L}_1} = \mathcal{L}_1(W \odot \hat{C}, W \odot C), \quad \mathcal{L}_{\text{reg-ms-ssim}} = \mathcal{L}_{\text{ms-ssim}}(W \odot \hat{C}, W \odot C), \quad (10)$$

for photometric accuracy and structural coherence, respectively. Notably, we utilize the regularized multi-scale differentiable SSIM loss $\mathcal{L}_{\text{reg-ms-ssim}}$ rather than the single-scale version $\mathcal{L}_{\text{reg-ssim}}$ [44] used in previous works [1] for a larger perceptual field.

In addition, to enforce the ordinal stability on depth maps [45, 46] for 3DGS, we first downsample the pseudo-depth map $\tilde{D}$ and the rendered depth map $\hat{D}$ to low-resolution variants $\tilde{D}^*$ and $\hat{D}^*$ of size $N \times N$ pixels using bilinear pooling, which suppresses high-frequency noise while preserving relative depth ordering. We then introduce a new depth ranking regularized loss that penalizes violations of ordinal relationships between every pair of pixels in the downsampled depth estimates:

$$\mathcal{L}_{\text{depth}} = \frac{1}{N^4} \sum_{x_i,y_i=1}^{N} \sum_{x_j,y_j=1}^{N} \max(-(\tilde{D}^*(x_i,y_i) - \tilde{D}^*(x_i,y_j))(\hat{D}^*(x_i,y_i) - \hat{D}^*(x_i,y_j)), 0). \quad (11)$$

This loss function maintains scale invariance and reduces dependence on fine-grained structural details by operating on the downsampled representations. Benefiting from this loss function, our approach achieves robust depth optimization even when initialized with imprecise pseudo-depth estimates, as demonstrated in Tab. 6.

The final loss function is given as:

$$\mathcal{L} = \lambda_{\mathcal{L}_1}\mathcal{L}_{\text{reg-}\mathcal{L}_1} + \lambda_{\text{ssim}}\mathcal{L}_{\text{reg-ms-ssim}} + \lambda_{\text{depth}}\mathcal{L}_{\text{depth}}, \quad (12)$$

where the parameters $\lambda_{\mathcal{L}_1}$, $\lambda_{\text{ssim}}$, and $\lambda_{\text{depth}}$ are utilized to balance the different loss components.

## 5 Experimental Results

In this section, we first describe the implementation details of our approach and the datasets we used. Then we evaluate the proposed method on both real-world and simulated underwater scenarios.

Table 1: Reconstruction performance on the SeaThru-NeRF [2] dataset. We report PSNR↑, SSIM↑, and LPIPS↓ scores for the four underwater scenes. The average FPS↑ and Training Time↓ on an RTX4090 are also provided. The best and second-best results are **bolded** and underlined, respectively.

| Methods | IUI3 Red Sea | | | Curaçao | | | J.G. Red Sea | | | Panama | | | Avg. FPS | Avg. Time |
|---|---|---|---|---|---|---|---|---|---|---|---|---|---|---|
| | PSNR | SSIM | LPIPS | PSNR | SSIM | LPIPS | PSNR | SSIM | LPIPS | PSNR | SSIM | LPIPS | | |
| ZipNeRF [4] | 16.937 | 0.474 | 0.412 | 19.956 | 0.442 | 0.421 | 19.022 | 0.349 | 0.483 | 19.012 | 0.349 | 0.482 | 0.17 | 5h |
| SeaThru-NeRF [2] | 26.755 | 0.826 | **0.168** | 30.959 | 0.915 | 0.133 | 23.282 | 0.876 | **0.111** | 31.276 | 0.937 | **0.071** | 0.68 | 2h |
| 3D-Gauss. [9] | 22.980 | 0.843 | 0.246 | 28.313 | 0.873 | 0.221 | 21.493 | 0.854 | 0.216 | 29.200 | 0.893 | 0.152 | 318 | 13min |
| WaterSplatting [1] | 29.840 | 0.889 | 0.203 | 32.203 | 0.948 | 0.116 | 24.741 | 0.892 | 0.116 | 31.616 | 0.942 | 0.080 | 169 | 6.9min |
| **Plenodium** | **30.275** | **0.895** | 0.205 | **34.120** | **0.953** | **0.110** | **25.058** | **0.896** | 0.121 | **32.435** | **0.946** | 0.074 | 249 | 7.0min |

Table 2: Restoration performance on our simulated dataset. We report PSNR↑, and SSIM↑ for each subset scenes. The best and second-best results are **bolded** and underlined, respectively.

| Methods | Beach | | | | | | Street | | | | | |
|---|---|---|---|---|---|---|---|---|---|---|---|---|
| | | PSNR | | | SSIM | | | PSNR | | | SSIM | |
| | easy | medium | hard | easy | medium | hard | easy | medium | hard | easy | medium | hard |
| *FOG* | | | | | | | | | | | | |
| SeaThru-NeRF [2] | **28.416** | 24.799 | 16.439 | 0.9228 | 0.8888 | 0.7856 | 27.677 | 23.562 | 17.730 | 0.8697 | 0.8286 | 0.7244 |
| WaterSplatting [1] | 17.725 | 16.724 | 15.492 | 0.8549 | 0.7728 | 0.7475 | 26.595 | 24.816 | 22.510 | 0.8924 | 0.8551 | **0.8003** |
| **Plenodium** | 26.372 | **26.248** | **25.275** | **0.9435** | **0.9259** | **0.9051** | **27.978** | **26.071** | **23.060** | **0.9107** | **0.8791** | 0.7982 |
| *WATER* | | | | | | | | | | | | |
| SeaThru-NeRF [2] | 25.938 | 19.500 | 16.495 | 0.9094 | 0.8590 | 0.8085 | 25.268 | 21.414 | 19.420 | 0.8533 | 0.7919 | 0.7467 |
| WaterSplatting [1] | 17.592 | 15.892 | 15.021 | 0.8198 | 0.7725 | 0.7450 | 24.936 | 22.120 | 20.205 | **0.8793** | **0.8240** | 0.7546 |
| **Plenodium** | **26.360** | **24.723** | **23.636** | **0.9406** | **0.9119** | **0.8867** | **25.731** | **22.392** | **20.285** | 0.8594 | 0.7983 | **0.7556** |

## 5.1 Implementation and Datasets

**Implementation details.** Our implementation is based on the Nerfstudio [16, 47]. After COLMAP initialization, we use the pseudo-depth Gaussian complementation with $\tau_w = 0.99$ and $\tau_{near} = 0.5$ to enrich the initial set of 3D Gaussians. During training, we empirically set $\lambda_{\mathcal{L}_1} = 0.8$, $\lambda_{ssim} = 0.2$, and $\lambda_{depth} = 5$ in Eqn. 12. The patch number $N$ is set to 16. The maximum degree of the SH coefficients for the medium and Gaussian primitives is set as 3. We accumulate the absolute gradient norms of $\mu$ for finer densification following [48]. For the reconstruction task, we train the model and render novel views using Eqn. 5, whereas for the restoration task, we reuse the learned parameters to generate de-scattered images via Eqn. 1, focusing exclusively on the objects. All the experiments are conducted on an RTX 4090 GPU.

**Seathru-NeRF dataset.** The SeaThru-Nerf dataset [2] includes four real underwater scenes: IUI3 Red Sea, Curaçao, Japanese Gardens Red Sea, and Panama. There are 29, 20, 20, and 18 images in each scene, respectively, where 25, 17, 17, and 15 images are used for training and the rest of the images are for testing. These images are captured in RAW format by a Nikon D850 SLR camera in underwater conditions. Following [1, 2], these images are downsampled to an averaged resolution of $900 \times 1400$ pixels and COLMAP [10, 11] is employed to determine the camera poses and produce sparse point clouds for the initialization of 3DGS-based methods.

**Our simulated dataset.** We utilize Blender to simulate a dataset with precise GT for restoration evaluation. The dataset includes two scenes (beach and street), each degraded by two types of media (fog [49] and water) at three incremental intensity levels (easy, medium, and hard), yielding 12 systematically structured subsets. Each subset contains 100 images at a resolution of $512 \times 512$ pixels. We split them evenly into 50 training and 50 testing samples to ensure a balanced evaluation. Compared to existing simulated datasets [1, 2], our dataset provides enhanced accuracy, scale, and diversity, enabling robust evaluation across different media types and degradation intensities. Notably, we achieve heterogeneous media modeling by controlling the `Density` parameter within the `Principled Volume`, making our dataset more representative of real-world underwater environments. To isolate restoration quality from exposure, we preprocess restored images by linearly scaling their intensity to align with the GT's mean luminance before computing evaluation metrics. Since Blender provides accurate camera poses, we use COLMAP solely to generate sparse point clouds for initializing the set of 3D Gaussians. *More details regarding dataset construction procedures and comprehensive dataset analysis are provided in Sec. B.*

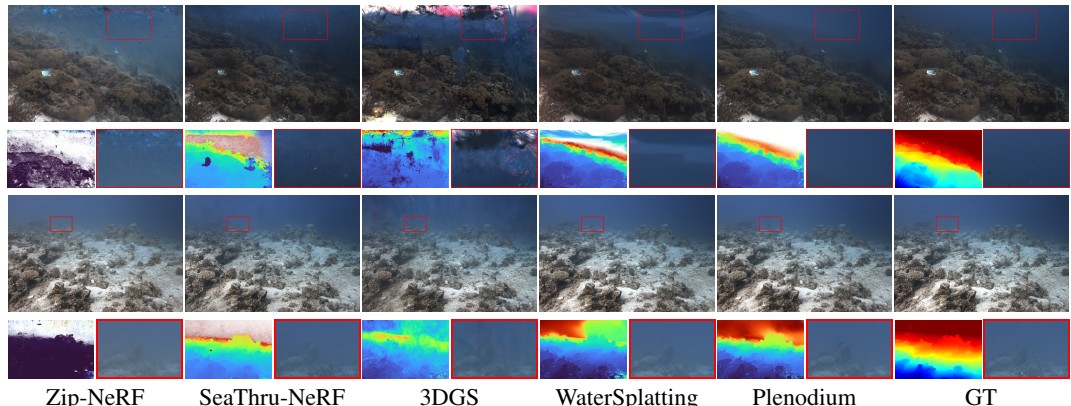

| Zip-NeRF | SeaThru-NeRF | 3DGS | WaterSplatting | Plenodium | GT |

Figure 3: Rendering performance comparison of our Plenodium against existing methods on the "Curaçao" and "IUI3 Red Sea" scenes. Rendered images and depths are presented for comparison. The pseudo-depth for ground truth is estimated using the Depth Anything Model [12, 13] for reference purposes. Compared to competing methods, Plenodium enhances clarity for medium and distant objects, as highlighted in the red boxes, while simultaneously yielding more reasonable depth maps.

Reconstruction Restoration Reconstruction Restoration Reconstruction Restoration

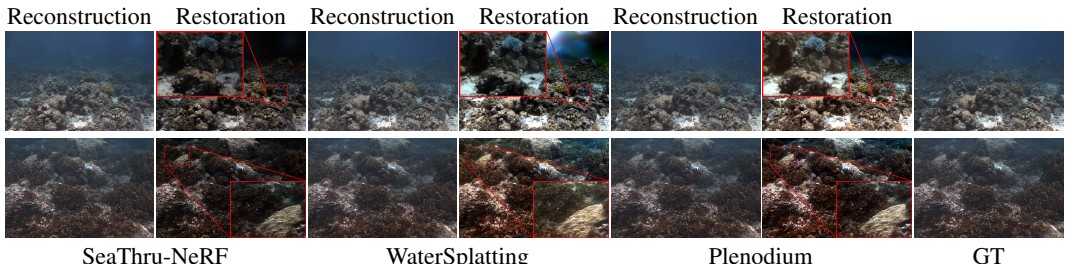

| SeaThru-NeRF | WaterSplatting | Plenodium | GT |

Figure 4: Restoration performance comparison of our Plenodium against existing methods on the "JapaneseGarden Red Sea" and "Panama" scenes. As shown in the red boxes, the proposed Plenodium generates results with more reasonable exposure and accurate colors.

## 5.2 Quantitative Results

We compare our approach against several state-of-the-art methods, including two NeRF-based ones (Zip-NeRF [4] and SeaThru-NeRF [2]) and two Gaussian splatting-based ones (3DGS [9] and WaterSplatting [1]). Note that SeaThru-NeRF and WaterSplatting are specifically tailored for underwater scene reconstruction. We use PSNR, SSIM, and LPIPS as metrics to evaluate the rendering quality. In addition, we benchmark the total training time and FPS (frames per second) on a machine with an RTX 4090 GPU to provide a comprehensive comparison of computational efficiency.

We first examine the 3D reconstruction performance of Plenodium on the SeaThru-NeRF dataset. As shown in Tab. 1, Plenodium performs favorably against state-of-the-art methods, increasing the PSNR and SSIM values by 0.872dB and 0.047 on average and improving the rendering speed by 47% (from 169FPS to 249FPS) over the best-competing method. Our approach substitutes the computationally expensive MLPs used in conventional methods with a lightweight SH-based spectral decomposition, significantly accelerating inference while preserving representational accuracy. We then evaluate the restoration performance of Plenodium on our simulated dataset with controllable scattering media in Tab. 2. Compared to state-of-the-art methods [2, 1], our Plenodium achieves better results, especially in challenging cases. This improvement stems from two key factors. First, benefitting from the proposed plenoptic medium representation, our approach is able to reconstruct 3D scenes more accurately. Second, the enhanced robustness of our approach originates from the integration of learned depth priors extracted from the Depth Anything Model, which supports: (i) pseudo-depth complementation to densify sparse points for initialization, and (ii) a ranking-based depth loss that enforces ordinal consistency during optimization. Taken together, these components allow Plenodium to achieve high-fidelity reconstructions under heterogeneous medium conditions and maintain reliable performance across varying levels of visual degradation.

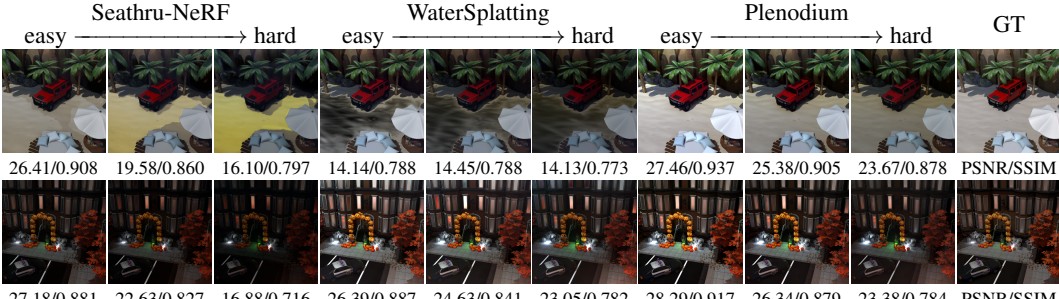

| Seathru-NeRF | | | WaterSplatting | | | Plenodium | | | GT |
| easy ⟶ hard | | | easy ⟶ hard | | | easy ⟶ hard | | | |

| 26.41/0.908 | 19.58/0.860 | 16.10/0.797 | 14.14/0.788 | 14.45/0.788 | 14.13/0.773 | 27.46/0.937 | 25.38/0.905 | 23.67/0.878 | PSNR/SSIM |
| 27.18/0.881 | 22.63/0.827 | 16.88/0.716 | 26.39/0.887 | 24.63/0.841 | 23.05/0.782 | 28.29/0.917 | 26.34/0.879 | 23.38/0.784 | PSNR/SSIM |

Figure 5: Restoration performance comparison of our Plenodium against existing methods on "Beach-Water" and "Street-Fog" scenes from our simulated dataset. Quantitative evaluations, including PSNR and SSIM metrics for each image, are beneath the corresponding figures. Among these methods, Proposed Plenodium produces results with more coherent textural details and superior color accuracy.

## 5.3 Qualitative Results

We provide a visual assessment of the 3D reconstruction capabilities of our proposed Plenodium against other state-of-the-art methods on the SeaThru-NeRF dataset in Fig. 3. Conventional methods, 3DGS and ZipNeRF, exhibit significant limitations when operating in scattering media, resulting in visually incoherent outputs marked by dense floaters and depth inconsistencies in underwater environments. SeaThru-NeRF and WaterSplatting show improvements by MLP-based medium modeling, thereby improving the rendering quality with enhanced photometric consistency. However, they still fail to accurately reconstruct fine details in water volumes and distant objects, as shown in the red boxes of Fig. 3. In contrast, our Plenodium yields visually superior results with clear water volumes and well-defined distant objects. As discussed in Sec. 6, our improved results stem from the proposed plenoptic medium representation, pseudo-depth Gaussian complementation, and depth ranking regularized loss. In addition, Plenodium produces the most accurate depth maps among the compared methods, validating its effectiveness in addressing the complexity of scattering media.

Due to the lack of de-watered GT in the SeaThru-NeRF dataset, we conduct a visual horizontal comparison of the de-water results across different methods. Fig. 4 presents qualitative comparisons on the "Japanese Garden Red Sea" and "Panama" scenes, where Plenodium achieves superior underwater restoration quality compared to other methods. Specifically, SeaThru-NeRF often produces underexposed images, leading to visually unappealing outputs with poor detail preservation. WaterSplatting introduces color distortions, such as greenish or yellowish tints on object surfaces, compromising the realism of restoration scenes. In contrast, Plenodium generates well-exposed outputs with effective medium separation, ensuring that submerged objects are rendered with natural color fidelity.

We further validate the restoration performance of Plenodium on our simulated dataset, which provides ground-truth, clear images for evaluation. Qualitative comparisons of the restoration performance on our simulated dataset are visualized in Fig. 5. SeaThru-NeRF suffers from severe color distortions, most prominently manifested as unnatural yellow discoloration of sand in "Beach-Water". It also frequently generates underexposed images, especially in the "Street-Fog" scene. WaterSplatting exhibits hollow reconstruction artifacts in the "Beach-Water" scene, due to COLMAP's failure under degraded visibility. Additionally, it retains residual fog on windows in the "Street-Fog" scene. Plenodium, by comparison, achieves more accurate color restoration and preserves fine textures, demonstrating greater robustness across diverse underwater scenes.

## 6 Analysis and Discussion

In this section, we present ablation studies to evaluate the contributions of key components in our framework, including the plenoptic medium representation, pseudo-depth Gaussian complementation, and the loss function design. *Additional discussion is provided in Sec. C.*

**Effect of the plenoptic medium representation.** To assess the effect of position information, as well as spherical harmonics (SH) encoding, in our plenoptic medium representation, we conduct an ablation study on the WaterSplatting framework, as shown in Tab. 3. We first compare the state-of-the-art WaterSplatting [1], which parameterizes the medium by an MLP with only the direction input

Table 3: Effect of the plenoptic medium presentation. All methods are evaluated on the SeaThru-NeRF [2] dataset.

| Method | PSNR | SSIM | LPIPS | FPS | Time |
|---|---|---|---|---|---|
| MLP w/ dir [1] | 29.600 | 0.918 | 0.129 | 179 | 6.9min |
| SH w/ dir | 30.124 | 0.920 | 0.129 | 265 | 6.0min |
| SH w/o dir&pos | 29.503 | 0.917 | 0.133 | 287 | 5.6min |
| SH w/ pos | 29.796 | 0.919 | 0.129 | 284 | 5.6min |
| SH w/ dir&pos | 30.254 | 0.921 | 0.127 | 257 | 6.0min |

Table 4: Results of our method w/o PDGC. Superscript values mark the differences in metrics when PDGC is disabled.

| Scene | init. #G | PSNR |
|---|---|---|
| IUI3 Red Sea | $21{,}907^{\,762\downarrow}$ | $30.176^{\,0.099\downarrow}$ |
| Curaçao | $25{,}837^{\,453\downarrow}$ | $33.996^{\,0.124\downarrow}$ |
| JapaneseGraden Red Sea | $21{,}140^{\,2{,}190\downarrow}$ | $24.947^{\,0.111\downarrow}$ |
| Panama | $22{,}501^{\,90\downarrow}$ | $32.434^{\,0.001\downarrow}$ |

Table 5: Effect of the PDGC.

| Initialization | PSNR | SSIM | LPIPS | FPS | Time |
|---|---|---|---|---|---|
| COLMAP | 30.388 | 0.9207 | 0.1274 | 238 | 7.0min |
| COLMAP + Rand. | 30.236 | 0.9218 | 0.1268 | 209 | 7.2min |
| COLMAP + PDGC | 30.472 | 0.9225 | 0.1276 | 249 | 7.0min |

Table 6: Effect of the loss functions.

| Loss Configuration | PSNR | SSIM | FPS | Time |
|---|---|---|---|---|
| $\mathcal{L}_{\text{reg-}\mathcal{L}_1} \& \mathcal{L}_{\text{reg-ssim}}$ | 30.307 | 0.9216 | 238 | 6.0min |
| $\mathcal{L}_{\text{reg-}\mathcal{L}_1} \& \mathcal{L}_{\text{reg-ssim}} \& \mathcal{L}_{\text{depth}}$ | 30.389 | 0.9204 | 253 | 6.1min |
| $\mathcal{L}_{\text{reg-}\mathcal{L}_1} \& \mathcal{L}_{\text{reg-ms-ssim}} \& \mathcal{L}_{\text{depth}}$ | 30.472 | 0.9225 | 249 | 7.0min |

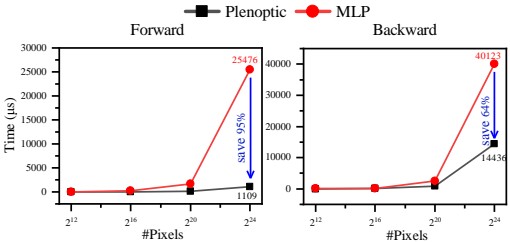

Figure 6: Efficiency comparison of our plenoptic medium representation against MLP-based representation in forward and backward.

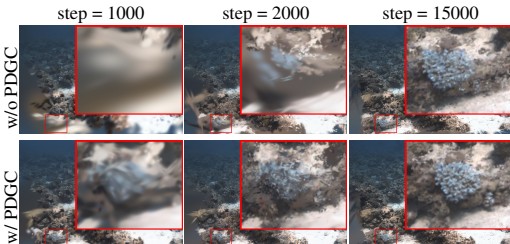

Figure 7: Rendering image comparison between w/ PDGC and w/o PDGC when different steps (1000, 2000, 15000).

(*i.e.*, MLP w/ dir), with a modified version that replaces the MLP with SH (*i.e.*, SH w/ dir). This substitution yields a 0.524 dB improvement in PSNR and an 86 FPS speedup, demonstrating SH's superior representational power and computational efficiency. To further investigate the contribution of position information, we begin with a shared SH parameterization across the entire volume using only degree-zero coefficients (*i.e.*, SH w/o dir&pos), which lacks both directional and spatial awareness. Building upon this, we introduce trilinear interpolation over the eight voxel corners to encode position (*i.e.*, SH w/ pos), leading to noticeable performance improvements. Finally, we combine the strengths of both directional and positional information, as well as SH encoding, in our plenoptic medium representation (*i.e.*, SH w/ dir&pos), which yields the highest rendering accuracy.

Furthermore, to validate the heterogeneous modeling capability of our plenoptic medium representation, we conducted a systematic investigation into the relationship between our learned SH coefficients and the physical properties of heterogeneous media using the "Beach-Fog-Hard" scene of our simulation dataset. Specifically, we examined the relationship between the learned SH coefficients for medium backscatter $A^{\sigma^{bs}}$ and the `Density` parameter set for the medium. As depicted in Fig. 8, our analysis reveals a statistically significant and strong correlation ($P < 0.0001$) between the learned SH coefficients and the ground-truth density values. This result provides compelling evidence that our representation is capable of modeling the complex, heterogeneous properties of scattering media.

**Efficiency of the plenoptic medium representation.** To assess the computational efficiency of our plenoptic medium representation compared to MLP-based representations, we quantified the forward and backward propagation latencies across varying pixel counts. The results, illustrated in Fig. 6, reveal that our plenoptic medium representation achieves significant speed improvements. During forward propagation, it requires only 5% of the time needed by the MLP baseline, while during backward propagation, it is less than half as time-consuming, which highlights the computational benefits of our plenoptic medium representation.

**Effect of the pseudo-depth Gaussian complementation.** To demonstrate the effectiveness of our pseudo-depth Gaussian complementation (PDGC) method, we remove PDGC and train this baseline using the same settings as ours. The quantitative results in Tab. 4 show that our PDGC supplements

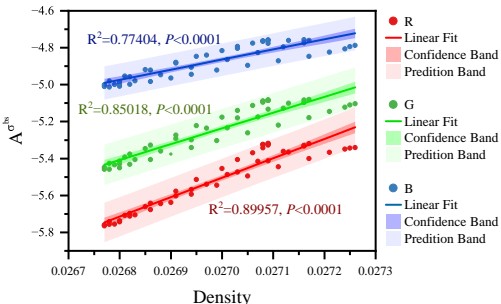
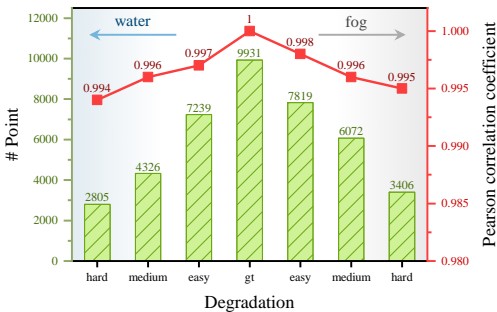

Figure 8: Correlation between learned coeffi-cients and medium physical properties.

Figure 9: Influence of degradation on COLMAP and depth estimation.

762, 453, 2190, and 90 Gaussian primitives for each test scene, improving the PSNR value by 0.099dB, 0.124dB, 0.111dB, and 0.001dB, respectively. To further analyze the effect of the proposed PDGC, we compare it with a baseline method that supplements Gaussians by randomly selecting positions rather than by our PDGC. The comparison results in Tab. 5 demonstrate the effectiveness of the proposed PDGC. Furthermore, Fig. 7 visually validates that the PDGC effectively improves the reliability of the initialization for 3DGS in degraded scenarios, producing high-quality reconstructions with significantly enhanced clarity compared to the baseline without using PDGC.

**Effect of the loss function.** To show the effect of our improved components in the loss function Eqn. 12, we compare with two baseline methods that respectively replace our loss with $\lambda_{\mathcal{L}_1}\mathcal{L}_{\text{reg-}\mathcal{L}_1} + \lambda_{\text{ssim}}\mathcal{L}_{\text{reg-ssim}}$ (which contains a single-scale structural similarity loss and is used in prior work [1]) or $\lambda_{\mathcal{L}_1}\mathcal{L}_{\text{reg-}\mathcal{L}_1} + \lambda_{\text{ssim}}\mathcal{L}_{\text{reg-ssim}} + \lambda_{\text{depth}}\mathcal{L}_{\text{depth}}$. Table 6 reveals that both the improved multi-scale SSIM loss and the proposed depth ranking regularized loss yield significant improvements in terms of PSNR and SSIM while imposing minimal additional computational overhead during training.

**Robustness of the pseudo-depth against degradation.** To quantitatively evaluate the robustness of the pseudo-depth estimation against various degradation levels, we plot the Pearson correlation coefficient in Fig. 9, which measures the agreement between pseudo-depth maps from degraded images and those from clean reference images on the "Street" scene of our simulated dataset. Simultaneously, we chart the number of 3D points estimated by COLMAP under different degradation conditions for comparison. As degradation intensity increases, COLMAP's point cloud density drops sharply, whereas the pseudo-depth consistently maintains high correlation values (e.g., $r > 0.99$), demonstrating its strong robustness and medium-agnostic properties. These results indicate that the depth prior is well-suited for initialization and supervision in challenging, degraded scenes.

# 7   Conclusion and Limitations

**Conclusion.** In this paper, we propose Plenodium, an efficient and robust framework for underwater 3D reconstruction. Our innovative plenoptic medium representation effectively integrates positional information to enhance medium modeling accuracy. Utilizing spherical harmonics-based encoding, we achieve a 47% speedup relative to WaterSplatting. Using pseudo-depth Gaussian complementation, we significantly improve the robustness of the initialization process. The proposed depth ranking regularized loss further improves the geometry by using depth order. Extensive evaluations in both real-world and simulated datasets with state-of-the-art methods demonstrate the effectiveness of our approach in reconstructing 3D underwater scenes and restoring underwater images.

**Limitation.** Despite the state-of-the-art performance achieved in underwater 3D reconstruction, the proposed Plenodium framework still faces several limitations. First, its perceptual consistency remains suboptimal. As shown in Tab. 1, Plenodium underperforms SeaThru-NeRF on the LPIPS metric in certain scenes, indicating that the reconstructed results may not always align well with human visual perception. Second, the water model in Eqn. 2 is inherently approximate and cannot fully capture the complex physical properties of heterogeneous media. This restricts our method to empirical fitting and limits its robustness and generalization in challenging environments.

## Acknowledgments

This work was supported by the National Natural Science Foundation of China (Grant Nos. 62332010 and 62272233).

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

## Summary of the Supplementary Materials

This appendix presents supplementary materials and analyses. We first present the implementation details of our method in Sec. A. We then detail the construction of our simulated dataset and analyze the effect of degradation on COLMAP initialization in Sec. B. In Sec. C, we conduct further ablation studies and provide an in-depth analysis of our findings. Section D provides additional qualitative visualizations to better illustrate the performance of our method. Finally, we consider our work's broader implications and potential societal impact in Sec. E.

## A   Implementation Detail

This section outlines our implementation details, including the training settings (i.e., Sec. A.1), supplementary information for pseudo-depth Gaussian complementation (i.e., Sec. A.2), and the depth gradient computation under scattering media (i.e., Sec. A.3).

### A.1   Training Settings

We train our model using a volumetric extension of 3D Gaussian Splatting. For reconstruction tasks, we train for 15,000 steps, while for restoration tasks, which require higher accuracy, we extend training to 30,000 steps. Following the progressive training strategy introduced in 3DGS [9], training begins at $1/4$ resolution and gradually doubles every 3,000 steps to increase spatial detail. To prevent unstable updates in the early training phase, we apply a 500-step warm-up before the Gaussian refinement. After warm-up, Gaussian refinement (including densification and culling) is performed every 100 steps. Densification is triggered for a Gaussian primitive when its gradient norm exceeds 0.0008. In this case, if the Gaussian scale is below 0.001, it is copied to expand coverage; otherwise, it is split into two samples to preserve fine-grained structure. In parallel, culling is applied at each refinement step to remove Gaussians with opacity below 0.5. To prevent opacity saturation and encourage stable convergence, all opacities are reset to 0.5 every five refinement steps. Together, these refinement steps first densify to improve coverage, then cull to remove floaters, ensuring a compact and effective representation.

Table 7: Optimizer and scheduler configurations for each parameter group.

| Parameter Group | Initial LR | Final LR | Notes |
|---|---|---|---|
| Means | 1.6e-4 | 5e-5 | Position updates |
| DC Features | 2.5e-3 | 2.5e-4 | Direct color channels |
| Rest Features | 1.25e-4 | 1.25e-5 | Non-DC channels |
| Opacities | 5.0e-2 | 5.0e-2 | No decay |
| Scales | 5.0e-3 | 5.0e-3 | No decay |
| Quaternions | 1.0e-3 | 1.0e-3 | Rotation parameters |
| Medium DC Features | 2.5e-3 | 2.5e-4 | For volumetric medium |
| Medium Rest Features | 1.25e-4 | 1.25e-5 | For anisotropic scattering |

We employ the Depth Anything Model [12, 13] as an external image depth estimator to generate the pseudo-depth maps. We use the latest version, V2, and the largest model variant, ViT-L, which is pretrained on diverse datasets and applied in inference mode without further fine-tuning. Following the official implementation, each image is first resized to a fixed resolution ($518 \times 518$) before passing through the model. This resizing ensures compatibility with the model's ViT backbone, which performs best under fixed input sizes due to its patch-based architecture. The predicted depth map is then upsampled via bilinear interpolation to match the original image resolution and stored as a dense pseudo-depth prior for further use in our pipeline.

Each parameter group is optimized using the Adam optimizer with $\epsilon = 10^{-15}$ and exponential decay scheduling. For instance, the 3D means are trained with an initial learning rate of $1.6 \times 10^{-4}$, which decays to $5 \times 10^{-5}$ over time, while opacities are optimized using a fixed learning rate of 0.05. Additional learning rates and scheduler configurations details are provided in Tab. 7.

### A.2   More Details of the Pseudo-Depth Gaussian Complementation

---

**Algorithm 1** Pseudo-Depth Gaussian Complementation

---

**Input:**
    The set of the input cameras $\mathcal{V}$, and corresponded images $\mathcal{C}$;
    The COLMAP initialized Gaussian primitives, $\mathbf{G}$;
**Output:**
    The final Gaussian primitives, $\mathbf{G}'$;
 1: $\mathbf{G}' = \emptyset$
 2: **for** $V \in \mathcal{V}, C \in \mathcal{C}$ **do**
 3:     $\hat{D}, T_{N+1}^{obj} \leftarrow$ render from $\mathbf{G}$ for $V$ using Eqn. 8.
 4:     $\Omega_p \leftarrow \{(x,y)|T_{N+1}^{obj}(x,y) \geq \tau_w\}$
 5:     get $\tilde{D}$ by Depth Anything Model with image input $C$
 6:     $\Omega_n \leftarrow \{(x,y)|\tilde{D}(x,y) < \tau_{near}\max(\tilde{D})\}$
 7:     get $\tilde{D}'$ from $\tilde{D}$ using Eqn. 9
 8:     **for** $(x,y) \in \Omega_n \cup \Omega_p$ **do**
 9:         get $\mu, A$ using Eqn. 13
10:         get $\Sigma$ using Eqn. 14
11:         $\sigma \leftarrow 0.1$
12:         $\mathcal{G} \leftarrow \{\mu, \Sigma, A, \sigma\}$
13:         $\mathbf{G}' \leftarrow \mathbf{G}' \cup \{\mathcal{G}\}$
14:     **end for**
15: **end for**
16: $\mathbf{G}' \leftarrow \mathbf{G} \cup \mathbf{G}'$
17: **return** $\mathbf{G}'$

---

In this section, we detail the procedure of our Pseudo-Depth Gaussian Complementation (PDGC), as summarized in Alg. 1.

Based on the pixel regions selected by $\Omega_n$ and $\Omega_p$ (as defined in Sec. 4.2), we determine where new Gaussians should be inserted. For each selected pixel $(x,y)$, we project it into 3D space as a Gaussian using its calibrated pseudo-depth $\tilde{D}'(x,y)$. The 3D mean position $\mu$ and the spherical harmonics-encoded color feature $A$ are computed as:

$$\mu = W^T \cdot \begin{bmatrix} \tilde{D}'(x,y) \cdot x \\ \tilde{D}'(x,y) \cdot y \\ \tilde{D}'(x,y) \end{bmatrix} + \begin{bmatrix} x_c \\ y_c \\ z_c \end{bmatrix}, \quad A = \text{RGB2SH}(C(x,y)), \quad \text{where } (x,y) \in \Omega_n \cap \Omega_p, \quad (13)$$

here, $W$ is the intrinsic matrix, and $[x_c, y_c, z_c]^T$ is the camera position. The function RGB2SH maps RGB values to 0th-order spherical harmonics coefficients for a compact color representation.

To represent the shape and orientation of each Gaussian, we define its covariance matrix $\Sigma$ via isotropic scaling $S$ and a random rotation $R$:

$$\Sigma = RSS^T R^T, \quad S = \text{diag}(s,s,s), \quad s = \frac{\tilde{D}'(x,y) \cdot (f_x + f_y)}{h + w}, \quad (14)$$

where $\text{diag}(s,s,s)$ constructs a diagonal matrix $S$ that uniformly scales the Gaussian along all three spatial axes, resulting in an isotropic shape. The scalar $s$ adapts the Gaussian size to the scene depth, while considering focal lengths $(f_x, f_y)$ and image dimensions $(h, w)$. The rotation matrix $R$ is randomly initialized to promote diversity in orientation and mitigate optimization bias.

### A.3 Backward Pass

Unlike standard 3DGS, where the depth $z_i$ mainly affects the rendered depth $\hat{D}$, in our medium-aware formulation, $z_i$ also influences the final rendered color $\hat{C}$ through scattering and attenuation. The loss $\mathcal{L}$ gradient concerning $z_i$ becomes:

$$\frac{\partial \mathcal{L}}{\partial z_i} = \frac{\partial \mathcal{L}}{\partial \hat{D}} \cdot \frac{\partial \hat{D}}{\partial z_i} + \frac{\partial \mathcal{L}}{\partial \hat{C}} \cdot \frac{\partial \hat{C}}{\partial z_i}. \quad (15)$$

The first term corresponds to the direct contribution of $z_i$ to the depth rendering, which follows the standard 3DGS formulation:

$$\frac{\partial \hat{D}}{\partial z_i} = \alpha_i T_i^{\text{obj}}.$$ (16)

The second term accounts for the influence of $z_i$ on color rendering, which stems from the medium-aware compositing process:

$$\hat{C} = \sum_i^N c_i \alpha_i T_i^{\text{obj}} e^{-\sigma^{\text{att}} z_i} + \sum_i^N c^{\text{med}} T_i^{\text{obj}} \left( e^{-\sigma^{\text{bs}} z_{i-1}} - e^{-\sigma^{\text{bs}} z_i} \right) + c^{\text{med}} T_{N+1}^{\text{obj}} e^{-\sigma^{\text{bs}} z_N}.$$ (17)

Thus, $z_i$ appears in the following terms of the color computation:

- $\hat{C}_i^{\text{obj}} = c_i \alpha_i T_i^{\text{obj}} e^{-\sigma^{\text{att}} z_i}$, where $z_i$ affects attenuation of the object.
- $\hat{C}_i^{\text{med}} = c^{\text{med}} T_i^{\text{obj}} (e^{-\sigma^{\text{bs}} z_{i-1}} - e^{-\sigma^{\text{bs}} z_i})$, where $z_i$ appears in the second exponential term.
- $\hat{C}_{i+1}^{\text{med}} = c^{\text{med}} T_{i+1}^{\text{obj}} (e^{-\sigma^{\text{bs}} z_i} - e^{-\sigma^{\text{bs}} z_{i+1}})$, where $z_i$ appears in the first exponential term.

Combining these, we get:

$$\frac{\partial \hat{C}}{\partial z_i} = -\sigma^{\text{att}} c_i \alpha_i T_i^{\text{obj}} e^{-\sigma^{\text{att}} z_i} + \sigma^{\text{bs}} c^{\text{med}} e^{-\sigma^{\text{bs}} z_i} \left( T_i^{\text{obj}} - T_{i+1}^{\text{obj}} \right),$$ (18)

where, the difference in transmittance simplifies as: $T_i^{\text{obj}} - T_{i+1}^{\text{obj}} = T_i^{\text{obj}} - (1 - \alpha_i) T_i^{\text{obj}} = \alpha_i T_i^{\text{obj}}$.

Then, we substitute it back into the gradient of the loss:

$$\frac{\partial \mathcal{L}}{\partial z_i} = \frac{\partial \mathcal{L}}{\partial \hat{D}} \cdot \alpha_i T_i^{obj} + \frac{\partial \mathcal{L}}{\partial \hat{C}} (\sigma^{bs} e^{-\sigma^{bs} z_i} c^{med} - \sigma^{att} e^{-\sigma^{att} z_i} c^{obj}) \cdot \alpha_i T_i^{obj},$$ (19)

This formulation captures depth's dual role in geometry and appearance, enabling more informative gradient flow in scattering environments.

## B  More Details of Our Simulated Dataset

In this section, we present additional details about our simulated dataset. We first describe the dataset construction process (i.e., Sec. B.1), including medium configurations and rendering settings. We then analyze the impact of different degradation levels on COLMAP-based initialization (i.e., Sec. B.2).

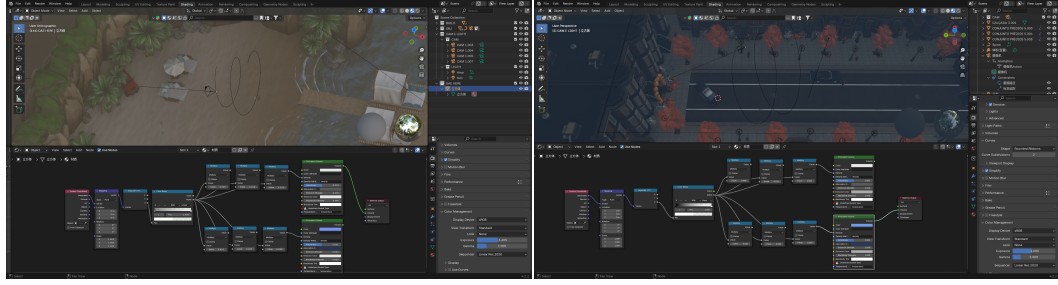

Figure 10:  Blender interface used for dataset rendering.

### B.1  Dataset Construction

As shown in Fig. 10, we simulate a scattering medium using Blender's `Principled Volume` shader, rendered with the `Cycles` engine to achieve high-fidelity light transport. A vertical density gradient is introduced along the Z-axis by combining the `Texture Coordinate`, `Mapping`, and `Separate XYZ` nodes, followed by a `ColorRamp` node to control the falloff. For fog, we adopt a white absorption color and low anisotropy (0.001) to simulate uniform scattering. For water, we use a bluish absorption

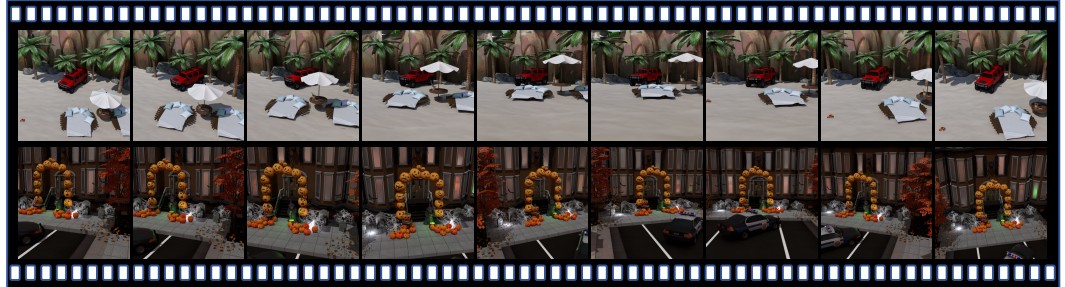

Figure 11: Sampled images from our simulated dataset.

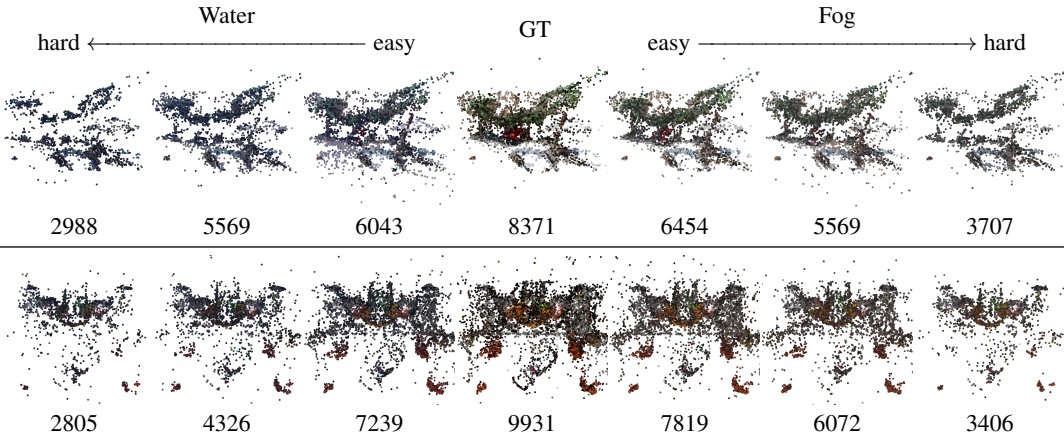

Figure 12: Sparse point clouds obtained by COLMAP under varying degradation conditions. The numbers below each image indicate the number of 3D points. For both the Beach (top) and Street (bottom) scenes, we show the impact of different levels of fog and water degradation (from easy to hard) compared to the clean ground truth. Severe degradation results in significantly sparser points, illustrating the challenge of reliable initialization of 3DGS.

tint and increased anisotropy to better approximate underwater light propagation with enhanced forward scattering. Three degradation levels (easy, medium, and hard) are realized by scaling the base density using adjustable `Multipliers` (e.g., 0.005, 0.01, 0.02). All images in our dataset are rendered with linear color management to allow for accurate exposure adjustments during post-processing. Specifically, we set the view transform to Standard and turn off gamma correction (gamma = 1.0). We do not use user-defined curve adjustments, ensuring no tone mapping or nonlinear operations alter the image. This enables consistent and physically meaningful exposure control during post-processing. The dataset comprises two distinct scenes (Beach and Street), as illustrated in the ground truth (GT) visualizations shown in Fig. 11, supporting robust and comprehensive benchmarking. Additional details, including exact shader setups and scene configurations, are provided in the supplementary Blender source files.

### B.2    Dataset Analysis

To evaluate the impact of image degradation caused by scattering media on the structure-from-motion (SfM) [10] initialization process in COLMAP [10, 11], we analyze the density and completeness of the generated sparse point clouds under degraded imaging conditions. When image quality is compromised due to fog or water, COLMAP struggles with reliable feature extraction and matching, resulting in significantly sparser and less accurate point clouds. As visualized in Fig. 12, specific regions, particularly those with strong scattering effects, exhibit apparent gaps or absences in the geometry. This degradation-induced sparsity directly hinders the quality of subsequent reconstruction stages, especially for methods relying on accurate geometry priors, such as 3DGS. These findings highlight the sensitivity of COLMAP-based initialization pipelines to visibility degradation, under-

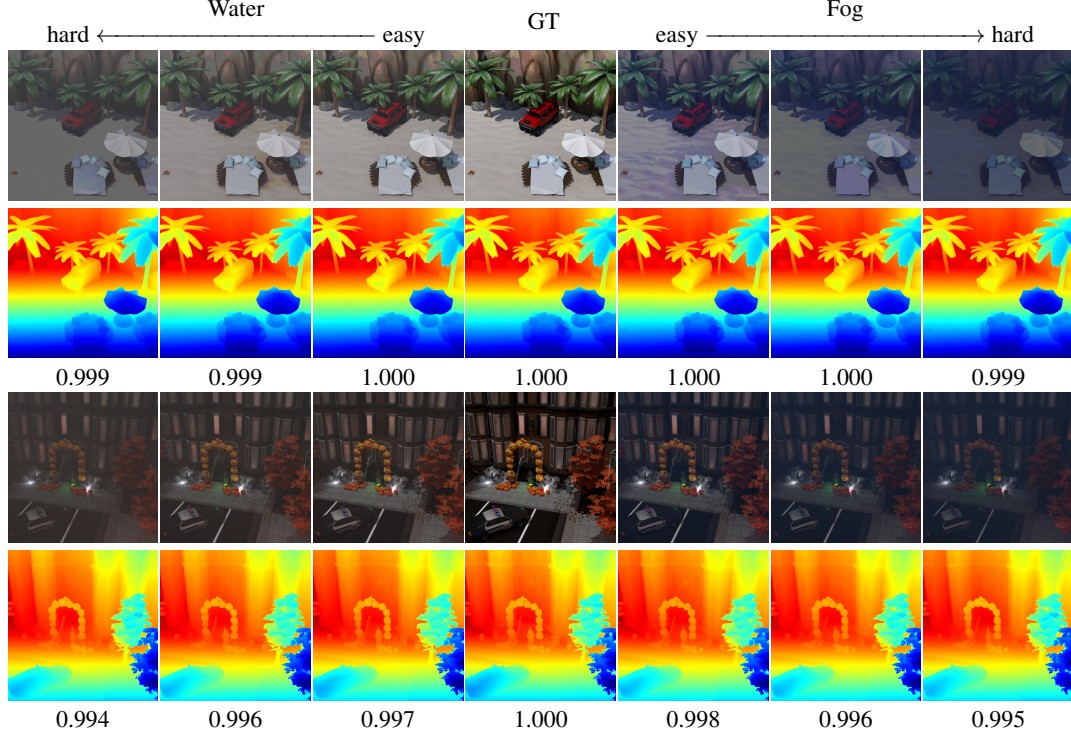

Figure 13: Pseudo-Depth estimated from various degraded images by [13]. Numbers below each map denote the Pearson correlation coefficient concerning the GT-based pseudo-depth. The consistently high values (close to 1.0) validate its effectiveness as a reliable depth in different environments.

scoring the need for complementary initialization strategies to recover missing geometry in severely degraded scenes.

## C  More Analysis and Discussion

In this section, we provide a comprehensive analysis of our method under various settings. We supplement a ColorChecker-based evaluation to verify the color fidelity of our approach for restoration (i.e., Sec. C.1) and assess the robustness of the pseudo-depth estimation under diverse degradation types (i.e., Sec. C.2). We then compare our Pseudo-Depth Gaussian Complementation (PDGC) with a graph-based densification strategy (i.e., Sec. C.3) and further analyze the effects of the regularization weight (i.e., Sec. C.4), COLMAP initialization (i.e., Sec. C.5), and depth ranking regularized loss (i.e., Sec. C.6). Furthermore, we examine how critical hyperparameters affect performance(i.e., Sec. C.7) and analyze statistical variance across different runs and degradation levels to establish result consistency (i.e., Sec. C.8). Finally, we analyze the limitations of our method with respect to the LPIPS metric (i.e., Sec. C.9).

### C.1  Colorchecker-based Evaluation

To validate the color fidelity of the proposed Plenodium, we supplement a ColorChecker-based evaluation. Following SeaThru [39], restored images are converted to a standard color space via the camera-pipeline manipulation platform [50], and white balance is set to an identity matrix derived from the Gray-World Hypothesis [51]. Color accuracy is quantified as the mean RGB angular error:

$$\bar{\psi} = \frac{1}{6} \sum_{i=1}^{6} \cos^{-1}(\frac{\|\hat{C}_i\|_1}{\sqrt{3}\|\hat{C}_i\|_2}), \tag{20}$$

between the six grayscale patches for each chart, following [34]. Table 8 demonstrates that our Plenodium attains the lowest angular error, further confirming the superior color fidelity. However,

Table 8: Comparison of RGB angular error across methods.

|  | SeaThru-NeRF | WaterSplatting | Plenodium |
|---|---|---|---|
| RGB angular error $\bar{\psi}$ | 13.039 | 11.527 | **8.288** |

Table 9: Effectiveness of the PDGC.

| Method | PSNR | SSIM | # G (init./final) |
|---|---|---|---|
| Plenodium $_{\text{w/o PDGC}}$ | 30.176 | 0.890 | 21,907 / 857,571 |
| Plenodium $_{\text{w/o PDGC \& w/ GU}}$ | 30.210 | 0.890 | 77,830 / 861,567 |
| **Plenodium (ours)** | **30.275** | **0.891** | 22,669 / 860,219 |

Table 10: Effectiveness of the regularized losses.

| Losses | PSNR | SSIM |
|---|---|---|
| $\mathcal{L}_{\mathcal{L}_1} + \mathcal{L}_{\text{ms-ssim}}$ | 29.35 | 0.892 |
| $\mathcal{L}_{\mathcal{L}_1} + \mathcal{L}_{\text{reg-ms-ssim}}$ | 29.60 | 0.910 |
| $\mathcal{L}_{\text{reg-}\mathcal{L}_1} + \mathcal{L}_{\text{reg-ms-ssim}}$ | 30.41 | **0.923** |
| $\mathcal{L}_{\text{reg-}\mathcal{L}_1} + \mathcal{L}_{\text{reg-ms-ssim}}$ | **30.47** | **0.923** |

we note that the SeaThru-NeRF dataset is not fully designed for color checker evaluation: only the "Curaçao" scene contains an unobstructed color chart, while the others suffer from occlusions or lack such references entirely, limiting the application of the color-checker-based metric.

## C.2 Robustness of Pseudo-Depth

Our pipeline leverages the Depth Anything Model [12, 13], a state-of-the-art monocular depth estimator, to compute robust pseudo-depth maps from media-degraded images. These maps serve as essential guidance for both our Pseudo-Depth Gaussian Complementation (PDCG) and the depth ranking regularized loss. As shown in Fig. 13, a key advantage of this approach is its robustness to medium-induced degradations. Despite varying levels of scattering and absorption in both water and fog, the pseudo-depth maps remain visually consistent across different input conditions and align well with those derived from clean ground-truth images. To quantitatively support this observation, we report the Pearson [52] correlation coefficient below each depth map, comparing each pseudo-depth to the one predicted from the clean (GT) image. The consistently high correlation values (e.g., >0.99) validate the robustness and medium-agnostic nature of the predictions by [13], making it well-suited for initialization and supervision in degraded scenes.

## C.3 Comparison Between PDGC and Graph-based Densification

We provide a comparison between our Pseudo-Depth Gaussian Complementation (PDGC) with a graph-based densification strategy (*i.e.*, Gaussian Unpooling, GU used in FSGS [52]). To quantitatively compare PDGC against GU, we conduct controlled experiments on the "IUI3-RedSea" scene and compare our Plenodium with two baselines that respectively remove our PDGC (i.e., Plenodium w/o PDGC) and replace our PDGC with GU (i.e., Plenodium w/o PDGC & w/ GU) for providing additional Gaussian during initialization while keeping all subsequent training settings identical. Compared to GU only using the information estimated by COLMAP, our PDGC further leverages an additional depth prior, thereby delivering superior quantitative results as illustrated in the Tab. 9. Moreover, GU introduces a large number of Gaussians at initialization, yet yields negligible gains in final reconstruction quality and leaves the final Gaussian count almost unchanged.

## C.4 Effect of the Regularized Losses

We further provide ablation evaluations on the effectiveness of the regularized in Eqn. 10 (i.e., the effectiveness of the weighting matrix $W = \frac{1}{\text{sg}(\hat{C})+\epsilon}$ on two losses in Eqn. 10). We compare our approach with the losses in Eqn. 10 (i.e., $\mathcal{L}_{\text{reg-}\mathcal{L}_1} + \mathcal{L}_{\text{reg-ms-ssim}}$) with three baseline methods that respectively remove the weighting matrix $W$ from the L1 loss (i.e., $\mathcal{L}_{\mathcal{L}_1} + \mathcal{L}_{\text{reg-ms-ssim}}$), remove the weighting matrix $W$ from the multi-scale differentiable SSIM loss (i.e., $\mathcal{L}_{\text{reg-}\mathcal{L}_1} + \mathcal{L}_{\text{ms-ssim}}$), remove the weighting matrix $W$ from both the L1 loss and the multi-scale differentiable SSIM loss (i.e., $\mathcal{L}_{\mathcal{L}_1} + \mathcal{L}_{\text{ms-ssim}}$). The comparison results in Tab. 10 show that our approach outperforms all the baselines, which demonstrates the effectiveness of emphasizing dark regions during optimization.

Table 11: Effect of the COLMAP initialization. We compare it with a random initialization with 50,000 points.

| Initialzation | PSNR | SSIM | LPIPS | FPS | Time |
|---|---|---|---|---|---|
| Random | 25.198 | 0.7983 | 0.2235 | 116 | 6.4min |
| COLMAP | 30.388 | 0.9207 | 0.1274 | 237 | 7.0min |
| COLMAP & PDGC | 30.472 | 0.9225 | 0.1276 | 249 | 7.0min |

Table 12: Effect of the depth ranking regularized loss. We compare it with the Pearson correlation loss from [52].

| Loss | PSNR | SSIM | LPIPS | FPS | Time |
|---|---|---|---|---|---|
| w/o $\mathcal{L}_{\text{depth}}$ | 30.305 | 0.9212 | 0.1272 | 252 | 7.0min |
| w/ $\mathcal{L}'_{\text{depth}}$ [52] | 30.384 | 0.9209 | 0.1292 | 246 | 7.7min |
| w/ $\mathcal{L}_{\text{depth}}$ | 30.472 | 0.9225 | 0.1276 | 249 | 7.0min |

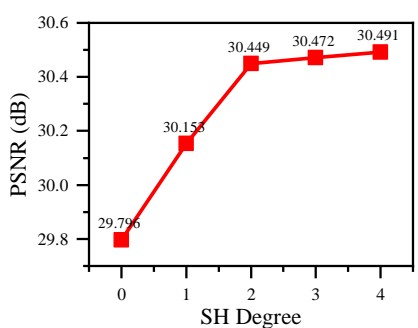

Figure 14: Effect of varying the maximum SH degree used for the plenoptic medium representation.

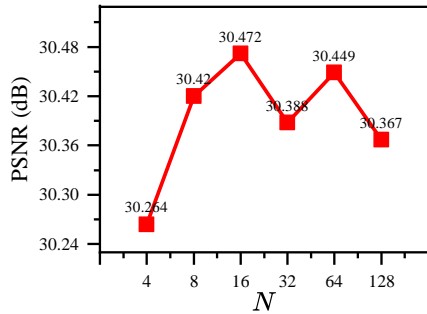

Figure 15: Effect of varying the number $N$ of patches used in the depth ranking regularized loss.

## C.5  Effect of the COLMAP Initialization

To evaluate the role of COLMAP-based initialization within our framework, we compare three variants: (1) our method with COLMAP initialization but without PDGC, (2) our method with random initialization using 50,000 uniformly sampled 3D points, and (3) our full pipeline combining COLMAP with PDGC. COLMAP provides a strong geometric prior that aids reconstruction; however, under severe degradation (e.g., fog or water), its output often becomes sparse and contains missing regions. In contrast, random initialization does not rely on scene-specific priors but ensures uniform spatial coverage, even in areas where COLMAP fails to generate points. As shown in Tab. 11, despite the degraded visibility, COLMAP initialization still leads to better performance than random initialization, validating the utility of its geometric prior. Moreover, our full method (augmenting COLMAP with PDGC, COLMAP & PDGC) further improves results, indicating that while COLMAP provides a solid foundation, complementary strategies can effectively enhance geometric priors under a degraded environment.

## C.6  Effect of the Depth Ranking Regularized Loss

To further evaluate the effectiveness of our proposed depth ranking regularized loss $\mathcal{L}_{\text{depth}}$, we compare our method (i.e., w/ $\mathcal{L}_{\text{detph}}$) against two baselines: one trained without any depth supervision (i.e., w/o $\mathcal{L}_{\text{depth}}$), and another using the Pearson correlation-based depth loss $\mathcal{L}'_{\text{depth}}$ adopted in FSGS [52] (i.e., w/ $\mathcal{L}'_{\text{depth}}$). As shown in Tab. 12, while $\mathcal{L}'_{\text{depth}}$ provides marginal improvements over the no-depth baseline, our method that leverages $\mathcal{L}_{\text{depth}}$ achieves superior performance, which shows that our depth ranking regularized loss offers more effective geometric supervision with imprecise pseudo-depth supervision.

## C.7  Effect of Hyperparameters

In our experiments, we investigate two critical hyperparameters that affect the performance of our plenoptic medium representation and the efficacy of the depth ranking regularized loss.

First, in Fig. 14, we control the maximum spherical harmonics (SH) degree for our plenoptic representation in our method. Adjusting this parameter determines the level of angular complexity captured in the medium field, thereby influencing the fidelity of volumetric effects such as scattering and color absorption. A higher maximum SH degree can model more detailed angular variations.

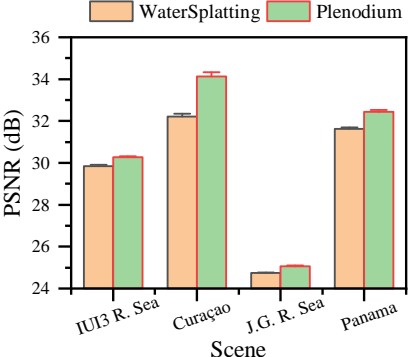
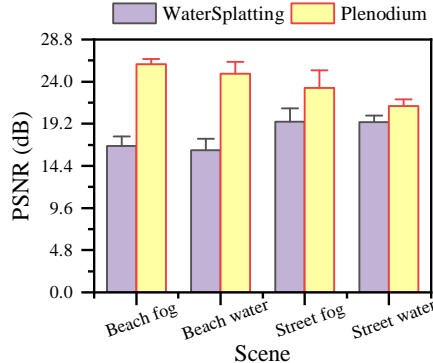

Figure 16: Mean and variance of reconstruction quality over four runs on real-world scenes.

Figure 17: Performance variation on simulated data across different degradation levels.

SeaThru-NeRF's LPIPS  Our LPIPS  GT  Noise in Background

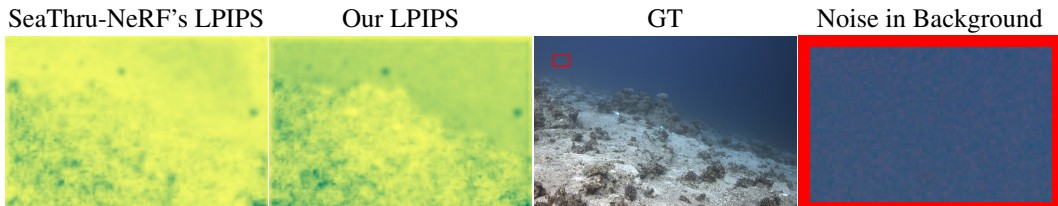

Figure 18: Visual comparison of LPIPS maps between SeaThru-NeRF and our Plenodium. The primary difference appears in the background regions corrupted by GT noise.

Still, it may also increase computational cost and risk of overfitting, whereas a lower degree results in a smoother but potentially oversimplified medium representation. To achieve an optimal trade-off between computational efficiency and representational fidelity, we fix the SH degree to 3.

Second, we vary the number $N$ of downsampled patches used in the depth ranking regularized loss for our method in Fig. 15. This loss plays a crucial role in enforcing depth consistency during training. A larger $N$ provides finer granularity for capturing local depth variations, but it also introduces more noise and increases computational overhead, even out-of-memory issues during training. In contrast, a smaller $N$ simplifies the loss calculation but may not capture sufficient spatial detail. Empirically, setting $N = 16$ yields the best performance while maintaining a reasonable computational load.

## C.8 Statistical Analysis

To ensure the robustness and stability of our quantitative results, we conduct four independent training runs on real-world scenes and report the average performance in Tab. 1 of the main manuscript. As shown in Fig. 16, we visualize the mean performance across runs and the corresponding variance to reflect consistency.

For our simulated dataset, we compare the average PSNR across scenes under water and fog degradation. As shown in Fig. 17, our method (Plenodium) consistently outperforms WaterSplatting across all conditions. The error bars represent the standard deviation across different degradation levels, reflecting both the effectiveness and robustness of each method under challenging visual environments.

## C.9 Limitation

While our method underperforms SeaThru-NeRF in terms of LPIPS in some scenes (as reported in Tab. 1 of the main manuscript), we conduct a visual analysis to better understand this discrepancy. As shown in Fig. 18, the LPIPS maps indicate that the main difference arises in the medium regions, where our method yields higher LPIPS values. We further observe that background areas in the GT contain visible noise, which may act as a confounding factor in LPIPS evaluation, limiting its reliability in degraded scenes.

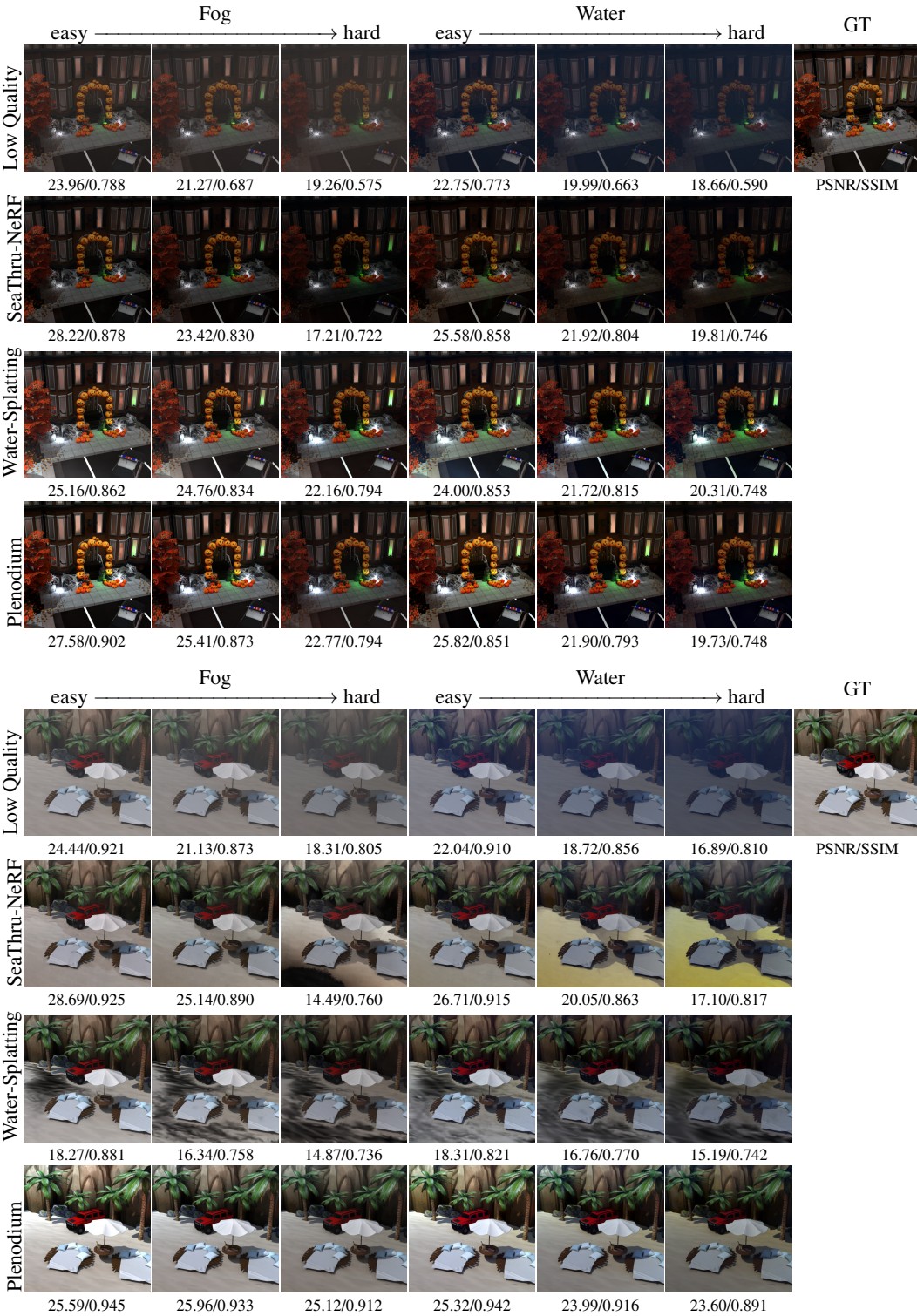

Figure 19: Visual comparison on our simulated dataset.

# D    More Visualizations

In this section, we present additional visualizations on our simulated dataset, comparing SeaThru-NeRF [2], WaterSplatting [1], and our proposed Plenodium, as shown in Fig. 19. We also include

video results in the supplementary material, rendered at 24 FPS using camera trajectories interpolated from the evaluation poses with a step size of 10.

# E  Broader Impact

Our method offers a more accurate and efficient solution for underwater 3D reconstruction, which can positively impact fields such as marine ecology, environmental monitoring, underwater archaeology, and infrastructure inspection. By improving scene recovery in visually degraded environments, our approach may assist in documenting underwater habitats, tracking pollution effects, and preserving submerged cultural heritage. Furthermore, the proposed simulated dataset provides a benchmark for evaluating underwater image restoration methods, promoting reproducibility and transparency. However, as with any enhanced visual sensing technology, there exists potential for misuse in surveillance or unauthorized mapping. We encourage responsible use and recommend that applications of this technology follow appropriate ethical and legal guidelines.

