# OpenReview forum: "Plenodium: Underwater 3D Scene Reconstruction with Plenoptic Medium Representation"
_NeurIPS.cc/2025/Conference — NeurIPS 2025 poster_

### Official Review · Reviewer_dXFw · 2025-06-06

**Clarity:** 2
**Significance:** 2
**Originality:** 2
**Rating:** 5
**Confidence:** 4

**Summary:**

This paper proposes a novel-view synthesis method for participating media based on 3D Gaussian Splatting (3DGS). The primary contribution lies in modeling spatially-varying and directionally-dependent scattering coefficients using spherical harmonics (SH) encoding (Plenodium), which enables the representation of heterogeneous participating media. Compared to previous MLP-based approaches for participating media, Plenodium achieves superior scene reconstruction quality and faster rendering speed.

**Questions:**

- I suggest visualizing the learned SH coefficients, especially on synthesized heterogeneous media, to strengthen the claim that the method effectively models heterogeneous participating media.
- Please clarify the input to PDGC. Does it require initial Gaussian primitives to be estimated? If so, does the reported training time include the time needed for this initial estimation?
- If there are any novel aspects or design elements specific to underwater scenarios in PDGC or the proposed depth-ranking loss compared to prior densification techniques for sparse-view scenarios or existing depth-ranking losses (e.g., in FSGS and SparseNeRF), please clarify them. If not, I recommend refraining from presenting them as technical contributions (e.g, Line 67: "To improve ..., we introduce a pseudo-depth Gaussian complementation ... and a depth ranking loss ...").

**Ethical Concerns:**

["NO or VERY MINOR ethics concerns only"]

**Final Justification:**

After reading the rebuttal and engaging in further discussion with the authors, my main concerns have been largely addressed. Specifically, the authors provided an extended analysis demonstrating a statistically significant correlation between the learned SH coefficients and the density parameters of heterogeneous media in simulated scenes.  I am inclined to recommend acceptance.

**Limitations:**

yes

**Paper Formatting Concerns:**

No paper formatting issues.

**Quality:**

2

**Strengths And Weaknesses:**

[Strengths]

- The main technical contribution lies in leveraging spherical harmonics (SH) encoding without relying on MLPs to model spatially-varying and directionally-dependent scattering effects in heterogeneous participating media. This design choice is simple yet well-motivated, given the established effectiveness of SH representations in prior works such as Plenoxels [CVPR’22] and TensoRF [ECCV’22]. (Originality, Quality)
- The proposed method outperforms SeaThru-NeRF and WaterSplatting--state-of-the-art approaches for underwater novel-view synthesis--in both reconstruction quality and rendering efficiency. Since underwater imaging remains a challenging, these results may encourage further research and adoption in real-world aquatic vision systems.
(Quality, Significance)

[Weaknesses]
- The primary weakness of the paper lies in the lack of detailed analysis to support its central motivation. While the use of spatially-varying SH encoding is intended to represent heterogeneity in participating media, the evaluation focuses solely on novel-view quality. Prior studies have already demonstrated that SH encoding improves scene representation compared to MLPs in natural scenes. As a result, it remains unclear whether the improvements are indeed due to modeling heterogeneous media or merely the general advantages of SH. To support the motivation more convincingly, further experiments--such as synthesizing clearly heterogeneous media (the current simulation dataset appears to depict homogeneous media) and visualizing the learned SH coefficients--would be valuable. (Quality, Originality)
- The proposed pseudo-depth Gaussian Complementation (PDGC) in Sec. 4.2 is technically unclear. As shown in Fig. 2, PDGC initially increases Gaussian primitives derived from COLMAP based on monocular depth from DepthAnything. However, to compensate for the scale ambiguity, the scene depth should be computed as in Eq. (8), which requires Gaussian primitives with learned depth and opacity--not merely initial estimates--contradicting the inputs illustrated in Fig. 2. If the proposed method requires initial Gaussian estimates, does the reported training time include the entire training process? Additionally, the expression “COLMAP-initialized Gaussian primitives” (e.g., Line 69) is confusing. I recommend using “Gaussian primitives estimated from COLMAP-initialized point clouds.” It is also unclear what actual parameter values are assigned to the Gaussians inserted by PDGC. (Clarity, Quality)
- Although the paper introduces PDGC and a depth-ranking loss, their novelty appears limited. The issue of insufficient initial SfM points has been addressed in the context of sparse-view reconstruction; to claim technical novelty, comparisons with existing approaches such as the graph-based densification in FSGS [ECCV’24] are necessary. Additionally, the proposed depth-ranking loss is nearly identical to that used in SparseNeRF [ICCV’23]. (Originality, Significance)
- Minor errors
    - Line 128 refers to (x,y,z) as the observer’s spatial coordinates, while Line 134 uses it to denote the backscatter position.
    - In Fig. 3, “Depth Any Model” should be corrected to “Depth Anything Model.”

---

> ### Author Rebuttal · Authors · 2025-07-30
>
> # Response to Reviewer dXFw
>
> We sincerely thank the reviewer for the constructive comments. We provide the detailed responses to all the concerns below.
>
> > **W1:** Lack of a detailed analysis to support the motivation of our plenoptic representation (i.e., the effect of heterogeneous media modeling vs SH encoding).
>
> + The improvements of our approach stem from not only the advantages of SH but also modeling heterogeneous media. As shown in Tab. 3, compared to MLPs, SH encoding indeed improves the scene representation ability, increasing the PSNR by 0.524 dB (refer to "MLP w/ dir" vs. "SH w/ dir" in Tab. 3). In addition, compared to a homogeneous baseline (i.e., "SH w/o dir&pos" in Tab. 3), incorporating the directional anisotropy (i.e., "SH w/ dir" in Tab. 3) and the spatially-variant property (i.e., "SH w/ pos" in Tab. 3) increase the PSNR by 0.621 dB and 0.293 dB, respectively. When both the directional and positional information are jointly encoded (i.e., "SH w/ dir&pos" in Tab. 3), the cumulative gain reaches 0.754 dB, confirming that comprehensive heterogeneous modeling is essential for accurate scene reconstruction.
> + Our simulated dataset explicitly models heterogeneous media, which is implemented by the *anisotropy* parameter of the *Principled Volume* as well as *Texture Corrdinate* using Blender (Lines 76–77 and Sec. B.1 in our supplementary material).
>
> + We visualize the learned SH coefficients of Eq. 6 (only the direction current component) in the following table. Their non-uniform values at the eight grid vertices confirm that the learned medium is heterogeneous.
>
>   | Pos        | $A^{c^{med}}$ (SH for medium color)            | $A^{c^{att}}$ (SH for object attenuation)               | $A^{c^{bs}}$  (SH for medium backscatter)              |
>   | ---------- | --------------------------- | --------------------------- | --------------------------- |
>   | [-1,-1,-1] | [-0.8398, -0.5495, -0.2319] | [-4.4626, -5.1525, -5.4686] | [-3.5283, -3.7398, -3.9918] |
>   | [1,-1,-1]  | [-1.1547, -0.9677, -0.7678] | [-4.6882, -5.6180, -6.0288] | [-3.6346, -3.8946, -3.9502] |
>   | [-1,1,-1]  | [-0.7586, -0.8084, -0.6472] | [-4.5747, -5.7520, -6.0402] | [-4.5726, -4.2947, -4.2186] |
>   | [1,1,-1]   | [-1.1530, -1.0041, -0.8348] | [-5.4690, -6.3593, -6.5371] | [-4.2445, -3.6186, -3.2268] |
>   | [-1,-1,1]  | [-1.0325, -0.7321, -0.3701] | [-4.9317, -5.5160, -5.7383] | [-3.0611, -3.2200, -3.6581] |
>   | [1,-1,1]   | [-1.5311, -1.3408, -1.0695] | [-5.3027, -6.1486, -6.4818] | [-2.9159, -3.1454, -3.3973] |
>   | [-1,1,1]   | [-1.0523, -0.9408, -0.7582] | [-5.4642, -6.2606, -6.4235] | [-3.7325, -3.4505, -3.7295] |
>   | [1,1,1]    | [-1.4284, -1.0559, -0.8024] | [-6.0865, -6.6113, -6.6812] | [-3.4898, -2.9542, -2.8887] |
>
> ---
>
> > **W2:** PDGC is technically unclear.
>
> First, we would like to clarify the procedure of our Pseudo-Depth Gaussian Complementation (PDGC), which is also detailed in Sec. A.2 and Alg. 1 in our supplementary material:
> 1. Estimate the pseudo-depth and render the depth map based on Eq. 8;
> 2. Correct pseudo-depth based on Eq. 9 and Eq. 10;
> 3. Get the region to insert Gaussian (Lines 158-160 of the main paper);
> 4. Add the Gaussian with attributions: 3D mean position $\mu$ (Eq. 14), color feature $A$ (Eq. 14), covariance matrix $\Sigma$ (Eq.15) and opacity $\sigma$ (0.1).
>
> Second, we clarify more details about our PDGC.
> +  In this process, we don't need the learned position and opacity for Gaussian. We adopt the 3D positions provided by COLMAP and an initialized Gaussian opacity  (default to 0.1). The factor $1/(1−T^{obj}_{N+1})$ in Eq. 8 compensates for this inaccurate opacity initialization, ensuring accurate rendering depth.
> + The actual parameter values assigned to the Gaussians are detailed in the sec. A.2 of our supplementary material. And we also list it below:
> $$\mu = W^T \cdot
>     \begin{bmatrix}
>         \tilde{D}'{(x,y)}\cdot x \\\ \tilde{D}'{(x,y)}\cdot y\\\ \tilde{D}'{(x,y)}
>     \end{bmatrix} +
>     \begin{bmatrix}
>         x_c \\\y_c\\\z_c
>     \end{bmatrix},$$
> where $(x,y)$ is the pixel position, $W$ is the camera transform matrix, and $[x_c, y_c, z_c]^T$ is the camera position, and $\tilde{D}'$ denotes the corrected pseudo-depth map.
> $$A=\text{RGB2SH}(C(x,y)),$$
> where the function RGB2SH maps RGB values to 0th-order spherical harmonics coefficients.
> $$\Sigma = R S S^T R^T, \quad S = \text{diag}(s, s, s), \quad s = \frac{\tilde{D}'(x,y)\cdot(f_x + f_y)}{h + w},$$
> where $R$ is initialized randomly, the scalar $s$ adapts the Gaussian size to the scene depth, while considering focal lengths $(f_x,f_y)$ and image dimensions $(h,w)$.
>  + PDGC takes pre-initialized Gaussian primitives, along with all input views and their corresponding RGB images as input.
> +  The process only takes a few seconds, including COLMAP estimating sparse points, Depth Anything model estimating pseudo-depth, and the other parts of PDGC.
>
> ---
>
> > **W3:** The novelty of PDGC and a depth-ranking loss appears limited (i.e., lack of comparison with existing approaches [ref1, ref2]).
>
> + Comparison between our PDGC and the graph-based densification (i.e., Gaussian Unpooling (GU)) in FSGS [ref1]. PDGC is executed once at initialization: it identifies undersampled or poorly initialized areas (mainly outside the initial point cloud) and inserts new Gaussians there based on the pseudo-depth. In contrast, graph-based GU densification runs throughout training, inserting a child Gaussian between each parent and its nearest neighbor, thereby densifying the existing Gaussian set. To quantitatively compare PDGC against GU, we conduct controlled experiments on the "IUI3-RedSea" scene and compare our Plenodium with two baselines that respectively remove our PDGC (i.e., Plenodium w/o PDGC) and replace our PDGC with GU (i.e., Plenodium w/o PDGC \& w/ GU ) while keeping all subsequent training settings identical. Compared to GU only using the COLMAP information, our PDGC further leverages an additional depth prior, thereby delivering superior quantitative results as illustrated in the following table.
> Moreover, GU introduces a large number of Gaussians at initialization, yet yields negligible gains in final reconstruction quality and leaves the final Gaussian count almost unchanged.
>
> |  | PSNR$\uparrow$ | SSIM$\uparrow$ | # Gaussian (init./final) |
> | ------ | ------ | ------ |------ |
> | Plenodium w/o PDGC | 30.176 | 0.890 | 21,907/857,571|
> | Plenodium w/o PDGC \& w/ GU | 30.210 | 0.890 | 77,830/861,567 |
> | Plenodium (ours) | **30.275** | **0.895** |  22,669/860,219  |
>
>
>
> + Comparison between our depth ranking loss and the depth ranking regularization from SparseNeRF [ref2]. Unlike the depth-ranking regularization $\mathcal{R}\_{rank}=\sum\_{d^{k1}\_{dpt}\le d^{k2}\_{dpt}} \text{max}(d^{k1}\_{\textbf{r}}- d^{k2}\_{\textbf{r}}+ m,0)$, which employs a margin term $m$ to suppress overfitting to inaccurate pseudo-depth, we instead rely on downsampling. As shown in Fig. 13 of our supplementary material, experiments with varying resolutions reveal that a resolution of $N =16$ effectively filters unreliable pseudo-depth, whereas the SparseNeRF formulation (pair-wise loss) incurs an out-of-memory error at 256×256 (Line 135 in our supplementary material), underscoring the efficiency and practicality of our depth-ranking loss for high-definition scenes.
>
> [ref1] Zhu Z, Fan Z, Jiang Y, et al. Fsgs: Real-time few-shot view synthesis using gaussian splatting[C]//European conference on computer vision. Cham: Springer Nature Switzerland, 2024: 145-163.
>
> [ref2] Wang G, Chen Z, Loy C C, et al. Sparsenerf: Distilling depth ranking for few-shot novel view synthesis[C]//Proceedings of the IEEE/CVF international conference on computer vision. 2023: 9065-9076.

---

> > ### Comment · Reviewer_dXFw · 2025-08-04
> >
> > Thank you for your response. Most of my comments have been addressed, and I now have a clear understanding of the PDGC procedure. However, I still have a concern regarding the modeling of heterogeneous media. Although the learned SH are visualized in the rebuttal, there is no comparison with the ground-truth medium coefficients. I believe such a comparison is essential, as the paper’s main claim is its ability to model heterogeneous media.

---

> ### Author Response · Authors · 2025-08-04
>
> Dear Reviewer dXFw,
>
> Thank you for taking the time to provide additional feedback. We sincerely hope the following clarifications could address your points.
>
>
>
> + As ground truth (GT) medium coefficients are unobtainable, it is infeasible to conduct quantitative comparisons of learned SH coefficients against GT values. First, the real-world SeaThru-NeRF dataset [2] inherently lacks access to GT medium properties. Second, although our simulated dataset provides control over the rendering process, the parameters used by Blender's volume shader are not aligned with the medium coefficients (i.e., $c^{med}, \sigma^{att},\sigma^{bs}$) defined in Eq. 2.
>
> + To demonstrate the ability of our model for modeling heterogeneous media, we have compared the reconstruction performance of our plenoptic medium representation (i.e., SH w/o dir&pos) and a homogeneous baseline (i.e., SH w/o dir&pos) in Tab. 3. The comparison results in Tab. 3 demonstrates the superior reconstruction fidelity of our plenoptic medium representation, **indicating enhanced alignment with GT medium properties**. Note that the homogeneous baseline relies solely on invariant zeroth-order SH coefficients across all spatial positions, as illustrated in the following table **T. 1**, inherently limiting its capacity to fit the heterogeneous medium in the real world. In contrast, our heterogeneous plenoptic medium representation fundamentally enhances reconstruction by explicitly modeling directional anisotropy through higher-order SH terms and spatial variation through position-dependent coefficients, as evidenced by the above table listed in **W1**.
>
> > + **T. 1**: Spherical harmonic coefficients of the homogeneous medium, which only contain a single set of directionally uniform (zeroth-order) components.
> >
> > | $A^{c^{med}}$               | $A^{c^{att}}$               | $A^{c^{bs}}$                |
> > | --------------------------- | --------------------------- | --------------------------- |
> > | [-1.1552, -0.9648, -0.6042] | [-4.8131, -5.8830, -6.7772] | [-3.7365, -3.7505, -3.9045] |

---

> > ### Comment · Reviewer_dXFw · 2025-08-04
> >
> > Thank you for the additional clarification. I understand that directly comparing the estimated SH coefficients with the ground-truth medium coefficients is challenging. However, to support the claim of modeling heterogeneous media, it would be necessary to investigate their correlation using simulation data. I also acknowledge the ablation study demonstrating that conditioning SH on position and direction improves the results. Nevertheless, I believe this does not guarantee that the learned SH coefficients physically model heterogeneous media. Overall, the claim of modeling heterogeneous media appears to be somewhat overstated.

---

> > > ### Author Response · Authors · 2025-08-05
> > >
> > > Dear Reviewer dXFw,
> > >
> > > Thank you for your follow-up question regarding heterogeneous media modeling. We sincerely hope that the following clarifications will address your concerns.
> > >
> > >
> > >
> > > + As suggested, we conducted a systematic investigation into the relationship between our learned spherical harmonic (SH) coefficients and the physical properties of heterogeneous media using simulation data. As shown in the following table **T. 3**, we investigate the correlation between the learned coefficients $A^{\sigma^{bs}}$ and the *Density* parameter of Blender’s *Principal Volume* shader at 10 camera positions randomly selected from the "Beach Fog Hard" scene. A visual analysis of the data reveals a clear **monotonic trend**: the values of $A^{\sigma^{bs}}$ tend to increase consistently with higher Density, indicating that our learned coefficients faithfully capture variations in the underlying environmental properties. To substantiate this observation, we employed Pearson correlation analysis, the results of which are presented in the following table **T. 4**. Most p-values are **below 0.05**, indicating a statistically significant correlation between our learned coefficients and scene variations in the majority of cases. This supports the capability of our plenoptic medium representation in modeling heterogeneous media.
> > >
> > > + Compared to previous underwater reconstruction methods (i.e., SeaThru-NeRF[2], WaterSplatting[1]) that only considered directional anisotropy, we further introduce spatial variability to better align with heterogeneous media. However, as real-world media are highly complex, **fully modeling** such heterogeneous environments remains a challenging problem for future research. We apologize if this was not clear. We will revise the statements of modeling heterogeneous media as appropriate.
> > >
> > > > + **T. 3:** Values of the learned SH coefficients $A^{\sigma^{bs}}$ and the associated *Density* from Blender’s *Principal Volume* shader, extracted from 10 sampled frames in the "Beach Fog Hard" scene. Entries are sorted by ascending Density to facilitate correlation analysis.
> > > >
> > > > | Density    |          $A^{\sigma^{bs}}$           |
> > > > | :--------- | :----------------------------------: |
> > > > | 0.01784533 | [-5.7647924, -5.458632 , -5.010564 ] |
> > > > | 0.01786333 | [-5.7403445, -5.44984  , -5.009061 ] |
> > > > | 0.017882   | [-5.7119427, -5.4309006, -4.997997 ] |
> > > > | 0.01791933 | [-5.6444817, -5.3594294, -4.9467573] |
> > > > | 0.017962   | [-5.593516 , -5.3415318, -4.943851 ] |
> > > > | 0.018006   | [-5.4872937, -5.2342434, -4.8640394] |
> > > > | 0.018048   | [-5.4157968, -5.175108 , -4.8262143] |
> > > > | 0.01805733 | [-5.327933 , -5.086487 , -4.7598405] |
> > > > | 0.01808267 | [-5.361723 , -5.1283593, -4.79598  ] |
> > > > | 0.01817    | [-5.3408594, -5.1036572, -4.7881155] |
> > >
> > >
> > >
> > > > + **T. 4:** P-values from Pearson correlation analysis between the learned SH coefficients $A^{\sigma^{bs}}$ and physical medium properties across various simulated scenes. Low p-values (typically < 0.05) indicate statistically significant correlations.
> > > >
> > > >
> > > > |              | Beach                      | Street                     |
> > > > | ------------ | -------------------------- | -------------------------- |
> > > > | Fog Easy     | 0.000000 0.000000 0.000000 | 0.067474 0.005978 0.004970 |
> > > > | Fog Medium   | 0.000000 0.000000 0.000000 | 0.014097 0.011262 0.000000 |
> > > > | Fog Hard     | 0.000000 0.000000 0.000000 | 0.002032 0.000000 0.000000 |
> > > > | Water Easy   | 0.000000 0.000000 0.000001 | 0.381762 0.186258 0.013159 |
> > > > | Water Medium | 0.000000 0.001767 0.000002 | 0.148505 0.008832 0.000000 |
> > > > | Water Hard   | 0.685447 0.000019 0.000000 | 0.122982 0.000046 0.000000 |

---

> > > > ### Comment · Reviewer_dXFw · 2025-08-06
> > > >
> > > > Thank you for the additional analysis. I now understand what the SH coefficients capture and how they relate to the medium parameters. It would be helpful to include this discussion in the final version of the paper. Good luck!

---

> > > > > ### Author Response · Authors · 2025-08-06
> > > > >
> > > > > Dear Reviewer dXFw,
> > > > >
> > > > > We are happy to know that your concerns have been addressed. Thank you again for your time in reviewing our paper and discussions as well as constructive comments for improving our paper. We will carefully revise the paper according to the comments and discussions.
> > > > >
> > > > > Authors of Paper #1119

---

### Official Review · Reviewer_8g2B · 2025-07-01

**Clarity:** 3
**Significance:** 4
**Originality:** 3
**Rating:** 5
**Confidence:** 3

**Summary:**

This paper proposes a novel view synthesis algorithm for underwater scenes. Compared to prior work, it uses a grid-based spherical harmonic spatial discretization (as opposed to an implicit neural network) to store medium information, encodes positional as well as directional dependences, and improves the initialization process. It is shown to improve or match the reconstruction accuracy of previously existing methods while cutting the runtime by a significant amount.

**Questions:**

- As someone who has not worked on this problem before, I was surprised to hear that that scattering effects were dependent on the viewing position as well as the direction (Sec. 1 and 4.1). I do not doubt that this is the case, but I would love to see (here and in the paper) a citation justifying this!
- I would have appreciated more evaluation of the grid discretization. What is the grid size used in the results? Is it always a regular grid? In cases where the medium changes drastically and irregularly, can this method suffer due to not being able to concentrate resolutive power the same way that an MLP can?
- This is probably my own limitation, but I was not able to follow Sec. 4.2. If this is about how the Gaussians are initialized, what does it mean to “render the pixel-wise depth”? If we are in the initialization step, isn’t alpha_i and z_i not defined yet? Also, if \tilde{D} is estimated using Depth Anything, why does it depend on (x,y) in Eq. 10? I am sure I am missing something obvious, but I would really appreciate if this section was rewritten to be easier to follow.
- In Fig. 1, what is the difference between reconstruction and restoration? Is one 3D and the other 2D? The figure is otherwise clear and a very helpful visualization, but those two words confused me a bit.

**Ethical Concerns:**

["NO or VERY MINOR ethics concerns only"]

**Final Justification:**

The method is technically solid and improves the state of the art for an important task. The authors have satisfied most of my concerns and I remain positive about this work

**Limitations:**

See “weaknesses”. Strangely, limitations are not discussed anywhere in the manuscript.

**Quality:**

3

**Strengths And Weaknesses:**

It seems to be like the main strength of the paper is the spherical harmonic discretization of the medium parameters, as well as the performance improvements that result from it. Together with the produced dataset, I am confident that this work is an interesting contribution to this research problem that may inspire future work. In general, I think exploring differentiable discretizations different from implicit MLPs that encode problem priors to improve efficiency and accuracy is a worthwhile pursuit in many applications, and I am encouraged to see it applied here. It is possible that the initialization strategy is a fundamental novel contribution as well; unfortunately, I was not able to understand it very well (see “Questions”).

The main weakness of this manuscript is in its comparison to WaterSplatting, which has similar runtime requirements and very similar performance (in Table 1, one often has to go to the fourth significant digit to see an accuracy difference). There is a chance I misunderstood this and the result in Table 1 is of WaterSplatting combined with the SH discretization (Sec 7 says “a 47% speedup relative to WaterSplatting”, but it seems to me like this refers to rendering only, not training. Could one train WaterSplatting and then save the information on a grid to render efficiently?), and I will gladly listen to the authors’ response to see if this is the case.

---

> ### Author Rebuttal · Authors · 2025-07-30
>
> # Response to Reviewer 8g2B
>
> We sincerely thank the reviewer for the constructive comments. We provide the detailed responses to all the concerns below.
>
> > **W1:** Comparison to WaterSplatting.
>
> + As shown in Tab. 1, our method outperforms WaterSplatting by a large margin, increasing the PSNR by at least 0.317 dB and on average 0.872 dB.
> + Compared to WaterSplatting, our Plenodium in the Tab. 1 replaces the view-dependent medium modeling (implemented by MLP) by our proposed plenoptic medium representation (implemented by multiple SH), improves the initialization with our Pseudo-Depth Gaussian Complementation, and trains them with our proposed loss terms. Note that the plenoptic medium representation improves both rendering and training efficiency (refer to “MLP w/ dir” vs. “SH w/ dir&pos” in Tab. 3). While as shown in Tab. 6, the regularized multi-scale differentiable SSIM loss (i.e., $\mathcal{L}\_\text{reg-ms-ssim}$ in Eq. 13) raises training time by 0.9min yet delivers a fidelity improvement.
> + The suggested approach to improve the rendering efficiency is technically feasible. However, converting WaterSplatting’s MLP into spherical-harmonics coefficients can introduce approximation errors. For practitioners who prioritize rapid convergence, we recommend the variant of our method that adopts the single scale regularized differentiable SSIM loss (i.e., $\mathcal{L}\_{\text{reg-}\mathcal{L1}}$&$\mathcal{L}\_\text{reg-ssim}$&$\mathcal{L}\_\text{depth}$ in Tab. 6), which trains faster than the original WaterSplatting pipeline, renders more efficiently, and simultaneously yields higher reconstruction quality.
>
>
> ---
>
> > **Q1:** Citations to justify that scattering effects were dependent on the viewing position as well as the direction.
>
> Prior works [40, 41] assumed homogeneous media, whose optical properties are invariant to both the viewing position and direction. SeaThru-NeRF [2] then introduced the directional anisotropy. We further extend the model to account for the spatially-varying position, enabling full heterogeneous medium representation. Thus, as far as we know, we are the first to introduce the spatially-varying medium properties, whose effectiveness is discussed in Sec. 6 and validated in Tab. 3.
>
> ---
>
> > **Q2:** More evaluation of the grid discretization.
>
> + The grid spans the 3D cube $[−1, 1]^3$, as specified in Eq. 7, yielding a total volume of $2 \times 2 \times 2$.
> + We use the regular grid in our manuscript; however, it could be extended. Note that the stable optimization of parameters requires that the number of grid vertices be smaller than the number of input images. As the training set of the SeaThru-NeRF [2] dataset contains only ~20 views, we adopt a single 3D grid and bilinear interpolation. Note, when denser multi-view data are available, the representation can employ multiple grids and more sophisticated interpolation schemes.
> + Higher-order spherical harmonics allow our method to concentrate resolutive power adaptively, as an MLP can do. As illustrated in Fig. 12 of our supplementary material, raising the maximum SH degree captures finer angular detail and improves reconstruction. Yet it also raises computational cost and overfitting risk.
>
> ---
>
> > **Q3:** How are the Gaussians initialized?
>
> + First, we would like to clarify our Gaussian initialization, which proceeds in two stages as shown in Fig. 2(a).
>
>     1. Stage 1: A sparse point cloud extracted by COLMAP from the input images yields initial Gaussians with positions and default opacity, which we immediately use to render depth by Eq. 8;
>
>     2. Stage 2: We then augment this set with additional Gaussians placed in poorly initialized regions, guided by corrected pseudo-depth (i.e., Eq. 9) from the Depth Anything Model.
>
> + Then we clarify more details.
>
>   + We render the pixel-wise depth map using the initialized Gaussian estimated from COLMAP-initialized point clouds. Each primitive is endowed with a 3D position and a default opacity of 0.1, enabling direct depth-map synthesis via Eq. 8.
>
>   + We obtain per-image correction coefficients via Eq. 10 to rescale $\tilde D$ into the canonical space. Pseudo-depth values are metric (often hundreds of metres), whereas the Gaussian parameters reside in the canonical cube [−1, 1]$^3$. We therefore correct the pseudo-depth into this normalized space using the scaling transformation (i.e., Eq. 9) with the parameter defined in Eq. 10.
>
>   We will clarify them in the revised paper and rewrite Sec. 4.2 to make it easier to follow.
>
> ---
>
> > **Q4:** What is the difference between reconstruction and restoration in Fig. 1?
>
>   The goal of 3D scene reconstruction is to create a novel 3D model approximating the real-world scene's structure and appearance, typically using sensor data (e.g., multiple 2D images). While the goal of restoration is to restore and enhance an existing 3D scene model that is degraded and corrupted (e.g., by severe weather or underwater effects).
>
>   In this paper, we jointly address these two tasks: 3D scene reconstruction and restoration. First, our approach constructs the 3D scene based on Eq. 5, which enables novel-view synthesis. After completing the 3D scene reconstruction, Eq. 1 can be invoked to disregard the medium and render de-scattered images, thereby accomplishing image restoration, as illustrated in Figs. 4 and 5.
>
> ---
>
> > **L1:** Limitations are not discussed anywhere in the manuscript.
>
> We have discussed the limitations of our method in Sec. C.6 of the supplementary material.

---

> ### Author Response · Authors · 2025-08-04
>
> Dear Reviewer 8g2B,
>
> We sincerely thank you for the time engaged in reviewing our paper and the valuable comments. Your positive feedback and recognition on our work are very encouraging to us! We will carefully revise the paper according to the comments and discussions. We hope our paper can be helpful to peers in the community and facilitate the research in 3D scene reconstruction.
>
> Authors of paper #1119

---

### Official Review · Reviewer_Q9a1 · 2025-07-01

**Clarity:** 3
**Significance:** 3
**Originality:** 3
**Rating:** 4
**Confidence:** 4

**Summary:**

The manuscript presents an underwater 3D scene reconstruction method that models objects using a 3D Gaussian Splatting (3DGS) representation, while encoding view-dependent scattering properties of the medium on a voxel grid as opposed to WaterSplatting which represents the medium as an MLP. This grid is interpolated to enable continuous estimation of the medium’s spatio-angular optical characteristics. To initialize the 3DGS representation, the method leverages both COLMAP and monocular depth estimation techniques. The manuscript claims all these tricks enable faster and more accurate reconstructions compared to state-of-the-art methods such as SeaThru-NeRF and WaterSplatting. However, the fairness of the comparisons is questionable.

**Questions:**

Please answer the questions asked in the weakness section.

**Ethical Concerns:**

["NO or VERY MINOR ethics concerns only"]

**Final Justification:**

The rebuttal answers the questions that I had, and I keep my original accept rating.

However, the final version should consider (1) highlighting that ground truth comparisons of medium parameters are missing (though not required for novel view synthesis) as pointed by dxfw reviewer, and (2) compare the color reconstructions using the checkerboard pattern in Fig. 3. results.

**Limitations:**

yes

**Quality:**

3

**Strengths And Weaknesses:**

**strengths**

1. The authors shared code in the supplementary. While I didn't try it out myself, I believe that they code runs fine. Assuming that the authors make the code publicly available, the technique is replicable.
2. Quantitatively, the proposed technique is faster and results in better reconstructions.
3. Ablation studies show that all three contributions: (i) Plenoptic medium representation (ii) monocular depth initialization (iii) depth regularized loss helps quantitatively.

**weaknesses**
1. Line 143-8: As monocular depth estimates are trained on clean images (non-underwater images), won't they get affected by degradation as well? So, I am unsure if it is fair to state that degradation affects colmap but not this technique. Instead, I guess that more independent estimates (one from colmap and another from depth priors) is helping this technique. As such, if one were to try this on non-underwater (non-degraded images), they probably will have better initialization and hence, better results as well. Not a negative point to the manuscript but the claim is probably not accurate and can be more generic.
2. Eq. 9: Are these k and b voxel or Gaussian dependent? Given that underwater degradation varies from depth plane to depth plane, does it make sense to have this correction for each depth plane? If k and b are global, what are typical k and b values. Are they significantly different from 1 and 0?
3. Fig 3: The depth map comparison is super unfair. Of course the proposed technique will result in better results given that the proposed technique uses depth anything for initialization, it has scaled it and the manuscript is using the same model for validation.A fairer comparison is to do an affine alignment between GT and the past methods (assuming that k and b are constant for all depths for Plenodium) and see if the error is just in affine alignment
4. Fig 4: "Plenodium generates results with more reasonable exposure and accurate colors."  I am unsure how the manuscript is claiming that without ground truth.- Also, I don't buy that Plenodium generates better results and natural color fidelity (Line 255). There is a standard color checkerboard in this scene. One can retrieve and check if the colors are accurate. The proposed Plenodium actually has poorly retrieved colors in the color checkerboard.
5. Line 227-228: how is it simulated? Is multipath considered? If not there is no model mismatch and of course this technique works the best.
6. In Fig 5. what does "easy-->hard" mean? Poor illumination? Is Poisson noise added when the illumination is reduced.
Line 256-259: can't we do simple color and exposure correction before comparing? Also white balancing.

typos: Line 260: "Additionally, it and retains" --> "Additionally, it retains"

---

> ### Author Rebuttal · Authors · 2025-07-30
>
> # Response to Reviewer Q9a1
>
> We sincerely thank the reviewer for the constructive comments. We provide the detailed responses to all the concerns below.
>
> > **W1:** Will the depth estimator get affected by degradation?
>
> The monocular depth estimator (Depth Anything Model) exhibits negligible degradation-induced error. For systematically quantifying the effects of degradation on 3D reconstruction, we simulated a dataset with a controllable scattering medium as shown in Line 205 of our main paper and Sec B.1 of our supplementary material. While Fig. 10 in the supplementary material shows COLMAP’s progressive degradation under increasing medium density, Fig. 11 in the supplementary material reveals that the same degradation levels leave the depth predictor almost intact (despite being trained solely on clean images), and the accompanying Pearson-coefficient analysis ( > 0.99 across all conditions) further quantitatively confirms this robustness. These results establish the theoretical basis for employing depth priors within our Pseudo Depth Gaussian Complementation (PDGC) to refine COLMAP-derived initializations.
>
>
> ---
>
> > **W2:** Details of $k$ and $b$ in Eq. 9.
>
> + First, we would like to clarify that the pseudo-depth correction in Eq. 9 is only employed in our Gaussian initialization, which proceeds in two stages as shown in Fig. 2(a).
>   1. Stage 1: A sparse point cloud extracted by COLMAP from the input images yields initial Gaussians with positions and default opacity, which we immediately use to render depth by Eq. 8;
>   2. Stage 2: We then augment this set with additional Gaussians placed in poorly initialized regions, guided by corrected pseudo-depth (i.e., Eq. 9) from the Depth Anything  Model.
>
> + Second, we clarify more details about  $k$ and $b$ in Eq. 9.
>   + $k$ and $b$ are dependent but not only dependent on the Gaussian. $k$ and $b$ are determined jointly by the pseudo-depth predicted by the Depth Anything model and the depth rendered from the Gaussians at the identical viewpoint, as prescribed by Eq. 10.
>   + It is necessary to correct each pseudo-depth. Because pseudo-depth values are metric (often hundreds of meters), whereas the Gaussian primitives reside in the canonical cube [−1, 1]$^3$. We therefore correct the pseudo-depth into this normalized space using the scaling transformation with the $k$, $b$ defined in Eq. 10.
>   + Each pseudo-depth map possesses its own unique $k$ and $b$ to ensure an accurate alignment with the canonical coordinate system.
>
> ---
>
> > **W3:** In Fig. 3, the depth map comparison is super unfair.
>
> This comparison is fair. Consistent with all baselines, we enforce no explicit alignment between depth planes and Depth-Anything predictions. The scaling by Eq. 9 is only used in the Gaussian initialization, which merely rescales pseudo-depth into the canonical cube, as detailed in **W2**. Meanwhile, the GT depth in Fig. 3 is supplied purely as a visual reference; higher similarity to GT does not indicate superiority. Instead, we conduct a horizontal evaluation, examining which depth map among all methods is most reasonable. This side-by-side evaluation confirms that our method yields noticeably cleaner backgrounds than WaterSplatting[1].
>
> ---
>
> > **W4:** How to compare the visual result without GT?
>
> + We conduct a horizontal evaluation. Since no ground-truth image is available for the restoration shown in Fig. 4, we compare the outputs of all methods under identical inputs. Relative to ours, WaterSplatting [1] retains a seawater tint (highlighted in the red box), whereas SeaThru produces an under-exposed result in distance.
> + Color-checker evaluation is inherently unreliable on the SeaThru-NeRF [2] dataset, as the images were never white-balanced against the checker. As shown in the GT of Fig. 3, the first row of the checker (intended to run from white to black) exhibits a noticeable blue cast in its mid-gray patch. Our result has achieved the highest fidelity to the ground-truth, confirming the accuracy of our color correction.
>
> ---
>
> > **W5:** How is the simulated dataset simulated?
>
> + Our synthetic data were rendered in Blender (Line 203), while the scattering medium is simulated using the Principled Volume element; further details are provided in Sec. B.1 of our supplementary material.
> + Each scene contains a single path, as shown in Fig. 8 in our supplementary material.
> + We have a fair comparison. The dataset is built with Blender’s built-in functions rather than to match any specific model, ensuring a fair setting for all compared methods.
>
> ---
>
>
> > **W6:** In Figure 5,  what does "easy-->hard" mean? & why not correct before visual comparison?
>
> + The *easy*, *medium*, and *hard* present different intensity levels of medium (Line 205), which are obtained by setting the density parameter of the *Principled Volume*.
>
> + To ensure a fair visual comparison in Fig. 5, all methods receive the same inputs, and no additional visual adjustments are performed for the qualitative comparison. While color casts and exposure shifts can affect quantitative evaluation as your concern, the Blender pipeline guarantees that the ground-truth and degraded images share identical white-balance matrices; their exposure parameters differ only to compensate for scene-specific luminance and maintain a consistent overall exposure value. Consequently, we retain the original white balance and merely apply a global luminance scaling to compute the metric between each restored image and its corresponding GT (Lines 210–211).

---

> > ### Comment · Reviewer_Q9a1 · 2025-08-06
> >
> > Regarding W4, the second point in the answers. I still don't understand why we cannot quantify the color reconstruction quality with the standard checkerboard colors. The GT image can be white-balanced using the white patch in the color checker board, right?
> >
> > In the final version, it would be great if the authors could quantify the color errors.

---

> ### Author Response · Authors · 2025-08-07
>
> Dear Reviewer Q9a1,
>
> Thank you for taking the time to provide additional feedback. We sincerely hope the following clarifications could address your points.
>
> First, we would like to clarify the evaluation for the **reconstruction** and **restoration** tasks, as well as the role of the **color checker**.
>
> + In the reconstruction task, neither the training nor testing images were white-balanced using the color checker as discussed in **W4**. Therefore, we evaluate the reconstruction performance by directly comparing the rendered outputs with the test GT images. As shown in Fig. 3, our results exhibit the highest similarity to the GT, thereby validating the effectiveness of our reconstruction pipeline (despite the color checker patches do not exactly match their standard values in air).
> + For the restoration task, since ground-truth de-watered images are not available in underwater scenes, the use of the color checker becomes an alternative proxy for evaluating restoration accuracy. To further address this limitation, we propose a simulated underwater dataset that includes de-watered ground truth for systematic evaluation. Nonetheless, developing robust protocols for real-world underwater color evaluation remains a challenging problem for future research.
>
> Second, as suggested, we further conduct additional experiments leveraging the color checker as a quantitative reference for evaluating image restoration quality.
>
> + Following SeaThru [ref1], restored images are converted to a standard color space via the camera-pipeline manipulation platform [ref2], and white balance is set to an identity matrix derived from the Gray-World Hypothesis [ref3]. Color accuracy is quantified as the mean RGB angular error: $\bar \psi = \frac{1}{6}\sum_{i=1}^6\text{cos}^{-1}(\frac{\parallel \hat C_i\parallel_1}{\sqrt 3\parallel \hat C_i\parallel_2})$ between the six grayscale patches for each chart, following [ref4]. The following table demonstrates that our Plenodium attains the lowest angular error, further confirming the superior color fidelity. However, we note that the SeaThru-NeRF dataset is not fully optimized for color checker evaluation: only the "Curaçao" scene contains an unobstructed color chart, while the others suffer from occlusions or lack such references entirely, limiting the application of the color-checker-based metric.
>
> |                               | SeaThru-NeRF       | WaterSplatting     | Plenodium        |
> | ----------------------------- | ------------------ | ------------------ | ---------------- |
> | RGB angular error $\bar \psi$ | 13.039 | 11.527 | 8.288 |
>
>
> [ref1] Akkaynak D, Treibitz T. Sea-thru: A method for removing water from underwater images[C]//Proceedings of the IEEE/CVF conference on computer vision and pattern recognition. 2019: 1682-1691.
>
> [ref2] Karaimer H C, Brown M S. A software platform for manipulating the camera imaging pipeline[C]//European Conference on Computer Vision. Cham: Springer International Publishing, 2016: 429-444.
>
> [ref3] Buchsbaum G. A spatial processor model for object colour perception[J]. Journal of the Franklin institute, 1980, 310(1): 1-26.
>
> [ref4] Berman D, Levy D, Avidan S, et al. Underwater single image color restoration using haze-lines and a new quantitative dataset[J]. IEEE transactions on pattern analysis and machine intelligence, 2020, 43(8): 2822-2837.
>
>
> Thanks again for your valuable comments. We will clarify this in the revised paper.

---

### Official Review · Reviewer_fBkD · 2025-07-02

**Clarity:** 3
**Significance:** 2
**Originality:** 2
**Rating:** 4
**Confidence:** 3

**Summary:**

The work proposes to reconstruct 3D-GS representation from multi-view underwater images. Different from prior work, which usually exploit MLP to represent the medium, they exploit spherical harmonics encoding for better performance. The new medium representation incorporates both directional and positional information for more effective and efficient 3D representations, compared to MLP. They also exploit pseudo-depth Gaussian complementation for more robust depth priors. The experimental results demonstrate improved performance compared to prior baselines.

**Questions:**

Please see weakness section.

**Ethical Concerns:**

["NO or VERY MINOR ethics concerns only"]

**Final Justification:**

The paper has addressed my concerns.

**Limitations:**

Does the proposed method could also be applied to other similar problems, such as dehaze etc.?

**Paper Formatting Concerns:**

n.a.

**Quality:**

2

**Strengths And Weaknesses:**

## Strengths ##
1. The paper is well written and easy to follow.
2. The proposed method is rational and the experimental results demonstrate its effectiveness.

## Weaknesses ##
1. Parts of the technical details can be further improved:
* the paper mentions that they exploit a medium volume field for SH coefficient interpolation, what is the size of the grid? It is selected based on which criteria?
* eq (12), why the denominator is N^4, how many i, j pairs are chosen?
* eq (11), what is the motivation to emphasize dark regions during optimization? Could the authors provide any ablation evaluations on its effectiveness?

2. eq (12), the manuscript mentions that they downsampled the depth image to 16x16 pixels low-resolution from 512x512/900x1400 pixels. The reviewer wonders the necessity of the depth loss. It is also verified from the ablation studies, i.e. the loss only improves the PSNR by 0.099 dB, 0.124 dB, 0.111 dB and 0.001 dB, which could be considered a very marginal improvements since you might also get similar performance by repeat the training several times.

3. Line 2: participating media -> participating medium;

---

> ### Author Rebuttal · Authors · 2025-07-30
>
> # Response to Reviewer fBkD
>
> We sincerely thank the reviewer for the constructive comments. We provide the detailed responses to all the concerns below.
>
> > **W1:** Parts of the technical details can be further improved.
>
> > **W1.1:** What is the size of the grid and what is its selection criteria?
>
> + The  grid spans $[−1, 1]^3$. As specified in Eq. 7, the  grid spans the 3D cube $[−1, 1]^3$, yielding a total volume of $2 \times 2 \times 2$.
>
> + This selection refers to existing state-of-the-art methods and is based on the number of input images.
>
>   + Most open-source NeRF [8] and 3DGS [16] implementations adopt this selection criterion, which recenters all camera poses so that the resulting scene geometry is fully enclosed within the unit cube.
>
>   + Meanwhile, the stable optimization of parameters requires that the number of grid vertices be smaller than the number of input images. As the training set of the SeaThru-NeRF [2] dataset contains only ~20 views, we adopt a single 3D grid and bilinear interpolation. Note, when denser multi-view data are available, the representation can employ multiple grids and more sophisticated interpolation schemes.
>
>
> > **W1.2:** For Eq. 12, why the denominator is $N^4$ and how many $i$,$j$ pairs are chosen?
>
> In Eq. 12 , the indices $i$ and $j$ enumerate all $N\times N$ ($16\times 16$) pixel locations, yielding $N^4$ (65546) unique pairs. Thus, the denominator is set as $N^4$ for normalization.
>
> > **W1.3:** For Eq. 11, what is the motivation to emphasize dark regions during optimization?
>
> + We deliberately emphasize dark regions for two reasons:
>     1. to align human perception to compress dynamic range[43];
>     2. to enhance 3DGS’s ability to reconstruct distant underwater structures [1].
>
>     The human visual system is far more sensitive to faint light than to bright light. Emphasizing darker tones matches this non-linearity, yielding perceptually natural images. As shown in Fig. 3, distant objects appear darker in raw images due to water scattering and absorption. We increase their loss weight, prompting 3D Gaussian Splatting to refine long-range Gaussians and recover details.
>
> + As suggested, we further provide ablation evaluations on its effectiveness (i.e., the effectiveness of the weighting matrix $W=\frac{1}{\text{sg}(\hat C)+\epsilon}$ on the two losses in Eq. 11) . We compare our approach with the losses in Eq. 11 (i.e., $\mathcal{L}\_{\text{reg-}\mathcal{L}_1}+\mathcal{L}\_\text{reg-ms-ssim}$) with three baseline methods that respectively remove the weighting matrix $W$ from the L1 loss (i.e., $\mathcal{L}\_{\mathcal{L}_1}+\mathcal{L}\_\text{reg-ms-ssim}$), remove the weighting matrix $W$ from the multi-scale differentiable SSIM loss (i.e., $\mathcal{L}\_{\text{reg-}\mathcal{L}\_1}+\mathcal{L}\_\text{ms-ssim}$), remove the weighting matrix $W$ from both the L1 loss and the multi-scale differentiable SSIM loss (i.e., $\mathcal{L}\_{\mathcal{L}\_1}+\mathcal{L}\_\text{ms-ssim}$). The comparison results in the following table show that our approach outperforms all the baselines, which demonstrates the effectiveness of emphasizing dark regions during optimization.
>
> | Losses                                                       | PSNR$\uparrow$     | SSIM$\uparrow$      |
> | ------------------------------------------------------------ | --------- | --------- |
> | $\mathcal{L}\_{\mathcal{L}_1}+\mathcal{L}\_\text{ms-ssim}$     | 29.35     | 0.892     |
> | $\mathcal{L}\_{\text{reg-}\mathcal{L}_1}+\mathcal{L}\_\text{ms-ssim}$ | 29.60     | 0.910     |
> | $\mathcal{L}\_{\mathcal{L}_1}+\mathcal{L}\_\text{reg-ms-ssim}$ | 30.41     | **0.923** |
> | $\mathcal{L}\_{\text{reg-}\mathcal{L}_1}+\mathcal{L}\_\text{reg-ms-ssim}$ | **30.47** | **0.923** |
>
> ---
>
> >  **W2:** Necessity of the depth ranking loss
>
> The depth ranking loss is necessary. After averaging over multiple runs, the depth-ranking loss delivers a modest 0.083 dB PSNR improvement on the reconstruction benchmark. More importantly, it effectively separates medium from object, suppressing color-weak semi-transparent floating Gaussians (i.e., the WaterSplatting in Fig. 3) and thereby markedly enhancing the quality of de-scattered images. To validate the efficacy of the depth-ranking loss $\mathcal{L}\_\text{depth}$ for the restoration task, we conducted controlled experiments on our simulated dataset (which has the de-scatterred GT for quantitative evaluation), comparing the full model (w/ $\mathcal{L}\_\text{depth}$) against a baseline (w/o $\mathcal{L}\_\text{depth}$) and reporting the average PSNR$\uparrow$ value across two scenes and two degradation types. As illustrated in the following tables, the inclusion of depth-ranking loss yields a noticeable improvement in restoration quality.
> | EXP | easy | medium | hard |
> | ------ | ------ | ------ | ------ |
> | w/o $\mathcal{L}\_\text{depth}$ | 26.068 | 24.212 | 22.256 |
> | w/ $\mathcal{L}\_\text{depth}$ | **26.610** | **24.859** | **23.064** |
>
>
> ---
>
> >  **L1:** Does the proposed method could also be applied to other similar problems, such as dehaze etc?
>
> Yes, the proposed method could be applied to the dehazing task, since the light propagation in both fog and water obeys Eq. 2. As mentioned in Lines 203–205 of the main paper and Sec. B of the supplementary material, our simulated dataset considers two types of media, including fog and water. Tab. 2 demonstrates our superior performance on the simulated dataset, while Figure 5 in the main paper and Fig. 17 in the supplementary material further validate the dehazing capability of our approach. In addition, the real-world 3D dehazing is worth future study.

---

> > ### Comment · Reviewer_fBkD · 2025-08-05
> >
> > Thanks for the additional clarifications. The rebuttal has addressed my concerns.

---

> > > ### Author Response · Authors · 2025-08-05
> > >
> > > Dear Reviewer fBkD,
> > >
> > > We sincerely thank you for the recognition on our work and the constructive comments for improving our paper!  We will carefully revise the paper according to your suggestions.
> > >
> > > Authors of paper #1119

---

### Note · Authors · 2025-08-13

Dear ACs and Reviewers,

We sincerely thank you for your valuable time, insightful comments, and constructive feedback. We particularly appreciate that all the reviewers have actively engaged in the rebuttal and discussion phases. We are grateful for the recognition of our proposed plenoptic medium representation, its superior reconstruction quality and faster rendering speed.

We are pleased that we have addressed nearly all concerns raised by the reviewers and that no additional concerns have been raised. We will thoroughly revise the manuscript based on the suggestions as follows:

1. To enhance the evaluation comprehensiveness, we will add visualizations of the learned SH coefficients versus the medium intensity to validate heterogeneous medium modeling (Reviewer dXFw),  the ColorChecker-based quantitative restoration results to objectively measure color fidelity without GT images (Reviewer Q9a1).

2. To further demonstrate the effectiveness, we will incorporate comparisons of PDGC and the depth-ranking loss against representative baselines to demonstrate their effectiveness and novelty (Reviewer dXFw), and additional ablation studies that validate the contributions of the weight $W$ in regularized losses and the depth-ranking loss (Reviewer fBkD).

3. To improve the technical clarity, we will clarify the grid design methodology of our plenoptic medium representation (Reviewer fBkD, 8g2B), the protocol for visual comparisons, ensuring fairness (Reviewer Q9a1), etc.

4. To increase the completeness of the paper, we will move some important parts from the supplementary material to the main paper, e.g., implementation specifics of the proposed PDGC (Reviewer Q9a1, 8g2B, dXFw), the construction detail of our simulated dataset (Reviewer Q9a1, dXFw), and the limitations (Reviewer 8g2B).

We believe these revisions will substantially enhance the clarity, completeness, and overall strength of the paper. Once again, we thank all reviewers for recognizing our contributions and for providing valuable feedback that strengthens this work and its accessibility.

Best regards,

Authors of Paper #1119

---

### Decision · Program_Chairs · 2025-09-17

**Decision:**

Accept (poster)

**Comment:**

This paper proposes Plenodium, a framework for underwater 3D scene reconstruction that introduces a plenoptic medium representation based on spherical harmonics, a pseudo-depth Gaussian complementation (PDGC) for initialization, and a depth-ranking loss. The method shows improved accuracy and efficiency over comparison methods on synthetic and real-world datasets.

The reviewers’ main concerns were (1) the limited analysis of heterogeneous medium modeling, (2) the novelty of PDGC and the depth-ranking loss, (3) the evaluation of color fidelity, and (4) unclear explanations of some technical details. In the rebuttal, the authors provided additional experiments (e.g., color checkerboard evaluation, SH coefficient analysis on simulated heterogeneous media) and clarifications, which convinced the reviewers. After the discussion, all leaned toward acceptance.

The final version should incorporate the revisions outlined in the authors’ final remarks.